# Controlling CRISPR-Cas9 with ligand-activated and ligand-deactivated sgRNAs

Kale Kundert [1], James E. Lucas[2], Kyle E. Watters [3], Christof Fellmann [3], Andrew H. Ng [2], Benjamin M. Heineike[4], Christina M. Fitzsimmons[5,11], Benjamin L. Oakes[3], Jiuxin Qu[6], Neha Prasad[6], Oren S. Rosenberg [6,7], David F. Savage[3], Hana El-Samad[7,8], Jennifer A. Doudna [3,9] & Tanja Kortemme [7,10]

The CRISPR-Cas9 system provides the ability to edit, repress, activate, or mark any gene (or DNA element) by pairing of a programmable single guide RNA (sgRNA) with a complementary sequence on the DNA target. Here we present a new method for small-molecule control of CRISPR-Cas9 function through insertion of RNA aptamers into the sgRNA. We show that CRISPR-Cas9-based gene repression (CRISPRi) can be either activated or deactivated in a dose-dependent fashion over a >10-fold dynamic range in response to two different small-molecule ligands. Since our system acts directly on each target-specific sgRNA, it enables new applications that require differential and opposing temporal control of multiple genes.

[1] Graduate Group in Biophysics, University of California San Francisco, San Francisco, CA 94158, USA. [2] UC Berkeley – UCSF Graduate Program in Bioengineering, University of California San Francisco, San Francisco, CA 94158, USA. [3] Department of Molecular and Cell Biology, University of California Berkeley, Berkeley, CA 94704, USA. [4] Bioinformatics Graduate Program, University of California San Francisco, San Francisco, CA 94158, USA. [5] Chemistry and Chemical Biology Graduate Program, University of California San Francisco, San Francisco, CA 94158, USA. [6] Department of Medicine, University of California San Francisco, San Francisco, CA 94158, USA. [7] Chan Zuckerberg Biohub, San Francisco, CA 94158, USA. [8] Department of Biochemistry and Biophysics, University of California San Francisco, San Francisco, CA 94158, USA. [9] Howard Hughes Medical Institute, University of California Berkeley, Berkeley, CA 94704, USA. [10] Department of Bioengineering and Therapeutic Sciences, University of California San Francisco, San Francisco, CA 94158, USA. [11] Present address: Laboratory of Cell Biology, Center for Cancer Research, National Cancer Institute, National Institutes of Health, Bethesda, MD 20892, USA. Correspondence and requests for materials should be addressed to K.K. (email: kale@thekunderts.net) or to T.K. (email: tanjakortemme@gmail.com)

CRISPR-Cas9 has emerged as an immensely powerful system for engineering and studying biology due to its ability to target virtually any DNA sequence via complementary base pairing with a programmable single-guide RNA (sgRNA)[1]. This ability has been harnessed to edit genomes, repress[2] or activate[3,4] gene expression, image DNA loci[5], generate targeted mutational diversity[6] and to modify epigenetic markers[7].

In addition to engineering CRISPR-Cas9 for diverse applications, there has also been broad interest in developing strategies to regulate CRISPR-Cas9 activity[8,9]. Such strategies promise to mitigate off-target effects and allow the study of complex biological perturbations that require temporal or spatial resolution[9]. To date, most of the progress in this area has been focused on switching the activity of the Cas9 protein using chemical[10–16] or optical[17–19] inputs. A general issue with these approaches is that all target genes are regulated in the same manner, although this limitation can be addressed with orthogonal CRISPR-Cas9 systems[16,20,21].

An alternative but less explored strategy is to regulate the sgRNA instead of the Cas9 protein. Since the sgRNA is specific for each target sequence, controlling the sgRNA directly has the potential to independently regulate each target. This strategy has been approached using sgRNAs that sequester the 20 nucleotide target sequence (the spacer) only in the absence of an RNA-binding ligand[22,23], ligand-dependent ribozymes that cause irreversible RNA cleavage[23,24], ligand-dependent protein regulators recruited to the sgRNA to alter CRISPR function[12], and engineered antisense RNA to sequester and inactivate the sgRNA[25].

Here we describe a new method to engineer ligand-responsive sgRNAs by using RNA aptamers to directly affect functional interactions between the sgRNA, Cas9, and the DNA target. In contrast to prior sgRNA-based methods, our approach can be used to both activate and deactivate CRISPR-Cas9 function in response to a small molecule. In addition, our approach requires only Cas9 and the designed sgRNAs. We further show that control of CRISPR-Cas9 function with our method is dose-dependent over a wide range of ligand concentrations and can be used to simultaneously execute different temporal programs for multiple genes within a single cell. We envision that this method will be broadly useful for many applications of CRISPR-Cas9-mediated biological engineering in bacterial systems.

## Results

**Design of controllable sgRNAs.** We sought to insert an aptamer into the sgRNA such that ligand binding to the aptamer would either activate or deactivate CRISPR-Cas9 function. We envisioned that ligand binding could either stabilize or destabilize a functional sgRNA conformation — bound to Cas9 and the DNA target — over other competing states in the ensemble (Fig. 1a). We chose the theophylline aptamer[26] as a starting point because it is well-characterized and has high affinity for its ligand, which is cell permeable and is not produced endogenously.

We first asked which sites in the sgRNA were most responsive to the insertion of the theophylline aptamer and which strategies for linking the aptamer to the sgRNA were most effective. We designed aptamer insertions at each of the sgRNA stem loops at sites that are solvent-exposed in the Cas9/sgRNA/DNA ternary complex[27] and exhibit various levels of tolerance to mutation[28]. These insertion sites are denoted the upper stem, nexus, and hairpin (Fig. 1b). We tested three linking strategies aimed at stabilizing a functional sgRNA conformation in the presence of the ligand: (i) replacing parts of each stem with the aptamer, (ii) splitting the sgRNA in half and using the aptamer to bring the halves together, and (iii) designing strand displacements (i.e. sequences that allow for alternative base pairing in the

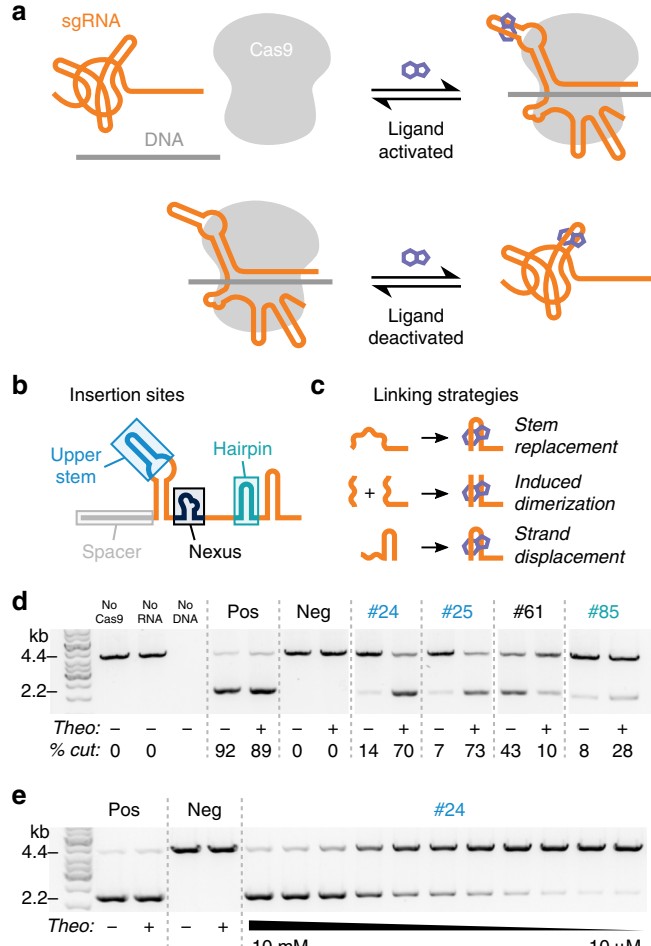

**Fig. 1** Design of ligand-controlled sgRNAs by inserting small molecule aptamers into the sgRNA. **a** Illustration of the design goal, where functional Cas9/sgRNA/target DNA complexes are stabilized either in the presence (top) or absence (bottom) of a small molecule ligand. **b** Aptamer insertion sites; sgRNA domains defined as in ref. 28. **c** Strategies for linking the aptamer to the sgRNA (Supplementary Table 1). **d** Efficiency of in vitro Cas9 cleavage of DNA in the presence and absence of 10 mM theophylline for controls and selected designs. Design numbers refer to Supplementary Table 1 and are color-coded by aptamer insertion sites defined in **b**. Percent cut values (bottom) are the average of at least two experiments. All data shown are from a single gel (some lanes are excluded for clarity). **e** Dose-dependence of cleavage efficiency in response to increasing concentrations of theophylline for a representative design (#24). Source data are available in the source data file

apo and holo states) (Fig. 1c, Supplementary Note 1, Supplementary Table 1).

To test the resulting 86 designed sgRNAs, we used an in vitro assay to measure differential Cas9-mediated DNA cleavage in the presence and absence of theophylline. We identified theophylline-responsive sgRNAs for all three insertion sites (Fig. 1d), with the most successful designs derived from the strand displacement linking strategy (Supplementary Table 1). We confirmed that the activity of our designs depended on the concentration of theophylline, as would be expected if the ligand affects function through binding the aptamer-containing designed sgRNA (Fig. 1e). In total, 10 designs were responsive to theophylline in vitro. For nine of these responsive designs theophylline addition activated CRISPR-Cas9 function, while for one design (#61) theophylline unexpectedly deactivated function.

Interestingly, all of the theophylline-activated designs had the aptamer inserted into either the upper stem or the hairpin, while the theophylline-deactivated design had the aptamer inserted into the nexus (Fig. 1b, d, Supplementary Table 1). These findings suggested the exciting possibility of regulating CRISPR-Cas9 function with both ligand-activated and ligand-deactivated sgRNAs, depending on the aptamer insertion site.

**Selection of controllable sgRNAs in *E. coli*.** We next sought to find designed sgRNAs that would function robustly in *E. coli*. To screen designed libraries using fluorescence-activated cell sorting (FACS), we changed to a cellular assay based on CRISPR-Cas9-mediated repression (CRISPRi) of super-folder green fluorescent protein (sfGFP) and monomeric red fluorescent protein (mRFP) (Fig. 2a)[14]. The strongest rational designs exhibited only weak activity in the CRISPRi assay (Supplementary Fig. 1). Since the sequences linking the aptamer to the remainder of the sgRNA affected activity in our in vitro experiments, we designed libraries with randomized linkers of 4–12 nucleotides at all three insertion sites to broadly sample different aptamer contexts (Fig. 2b, Supplementary Fig. 2, Supplementary Note 2, Supplementary Table 2). To identify ligand-activated sgRNAs, we screened each library first for CRISPRi activity in the presence of theophylline, then second for lack of activity in the absence of theophylline. We then repeated that selection/counter-selection with a different spacer to avoid selecting sgRNA scaffold sequences that would be specific for a particular spacer (Fig. 2c). To identify ligand-inhibited sgRNAs, we used an analogous four-step selection/counter-selection protocol that began by screening for activity in the absence of ligand. We validated the activity of the selected hits with a third spacer that was not used in any of the screens (Supplementary Table 3). The most robust ligand-activated sgRNA variant (termed ligRNA$^+$; i.e. sgRNA that is active in the + ligand state) and ligand-inactivated sgRNA (termed ligRNA$^-$) showed 11× and 13× dynamic ranges that spanned 55 and 59% of the range achieved by the controls, with negligible overlap between the active and inactive populations (Fig. 2d, e, Supplementary Table 4).

**Models for ligRNA mechanism.** The ligRNA$^+$ construct derived from inserting the aptamer into the hairpin while randomizing the remainder of the hairpin, the 5 unpaired nucleotides at the apex of the nexus, and the region between the hairpin and the nexus. The hairpin became more GC-rich, but the base-pairing was conserved with the exception of a single mismatch. The apex of the nexus became complementary to the 5′ region of the aptamer. The region between the hairpin and the nexus remained AU-rich and unpaired (Supplementary Fig. 3a). Secondary structure predictions of ligRNA$^+$ using ViennaRNA[29] (Supplementary Fig. 3b) are consistent with our intended mechanism, where ligand binding to the aptamer leads to strand displacements stabilizing the active sgRNA conformation.

The ligRNA$^-$ construct derived from inserting the aptamer into the nexus while randomizing the nexus stem. The nexus stem was extended from 2 to 5 bp, but remained base-paired and GC-rich (Supplementary Fig. 3a). The mechanism underlying the ability of ligRNA$^-$ to deactivate CRISPR-Cas9 function upon the addition of theophylline was unclear. ViennaRNA predictions of the lowest energy conformation for ligRNA$^-$ were uninformative on the mechanism of ligand control, as they suggested that ligRNA$^-$ adopts the same secondary structure in the presence and absence of theophylline (Supplementary Fig. 3c). However, we noticed that in the stem sequence selected in our screen, U95 in ligRNA$^-$ (U59 in the crystal structure of the Cas9 ternary complex with DNA and RNA[27]) was conserved in 17 of the 20

isolated sequences (Supplementary Table 3). This uracil makes specific hydrogen-bonding interactions with asparagine 77 of Cas9 in the ternary complex. In the sgRNA scaffold this uracil is always unpaired, but in ligRNA$^-$ it is predicted to engage in a wobble base pair with G65 in the stem leading up to the aptamer (Supplementary Fig. 3c). These observations led us to hypothesize that ligand binding to the aptamer controls the extent to which U95 is unpaired, which in turn determines whether or not ligRNA$^-$ interacts functionally with Cas9 and the target DNA. To test this hypothesis, we first designed strand-swapping mutations in the stem leading up to the aptamer (Supplementary Fig. 4). As expected, swapping U95 rendered ligRNA$^-$ completely inactive, while swapping base pairs at the positions between U95 and the aptamer had only a mild effect (Supplementary Fig. 4d). We then modulated the strength of the base pairs between U95 and the aptamer. Consistent with the hypothesis that ligand binding to the aptamer decreases access to U95, we found that weaker base pairs were more repressing while stronger base pairs were more activating (Supplementary Fig. 4e). These results provide a possible explanation for how ligRNA$^-$ deactivates CRISPR-Cas9 function in the presence of the ligand and suggest additional ways for ligRNAs to be tuned for specific applications.

**Tunability of ligRNA function.** Next, we tested whether the ligRNAs responded to increasing concentrations of theophylline in a dose-dependent manner in the cellular CRISPRi assay. We observed that the activities of both ligRNA$^+$ and ligRNA$^-$ were smoothly titratable and exhibited a nearly linear response over a large range of ligand concentration (Fig. 2f). We note that the apparent EC$_{50}$s for ligRNA$^+$ and ligRNA$^-$ (134.8 ± 11.3 μM and 177.6 ± 17.3 μM) are much higher than the $K_D$ of the theophylline aptamer alone (320 nM)[30]. This discrepancy is common for RNA devices[31] and could be explained by the altered structural context of the aptamer embedded in an sgRNA sequence. Nevertheless, the linear dependence on theophylline concentration of the ligRNAs demonstrates their utility for not only turning genes on or off, but also for precisely tuning their levels of expression.

A recent study reported regulation of CRISPRi activity by modulating sgRNA expression levels in *E. coli*[32]. To test how decreased expression levels would affect the ligRNAs, we replaced the strong constitutive promoter driving sgRNA expression (J23119) with a weak constitutive promoter (J23150), confirmed decreased sgRNA expression by quantitative real-time polymerase chain reaction (qPCR) (Supplementary Fig. 5a), and repeated the CRISPRi assay. Both ligRNAs remained functional with the weak promoter, albeit with somewhat narrower dynamic ranges (from a 10.2 ± 0.7-fold to a 5.9 ± 0.5-fold change upon theophylline addition for ligRNA$^+$ and from a 16.2 ± 1.0-fold to a 11.6 ± 0.9-fold change for ligRNA$^-$, Supplementary Fig. 5b). Notably, the weak promoter shifted the dynamic ranges of both ligRNAs in the direction of increased gene expression, to the point where nearly full gene activation was achieved in the non-repressing state. These results suggest that the ligRNAs are able to repress at low expression levels, and that tuning promoter strength is useful for applications that require full gene activation (alternatively, a collection of ligRNA variants that shift the dynamic range is shown in Supplementary Fig. 6).

**Rapid response time of ligRNAs to added ligand.** To determine the timescale of the response to ligRNA switching, we measured GFP mRNA levels by qPCR at timepoints ranging from 2 to 30 min after addition or removal of theophylline from the growth medium (Fig. 2g). Both ligRNAs effected a rapid change in mRNA levels. With ligRNA$^+$, the response was 50% complete after 10.2 min when adding theophylline and 4.8 min when

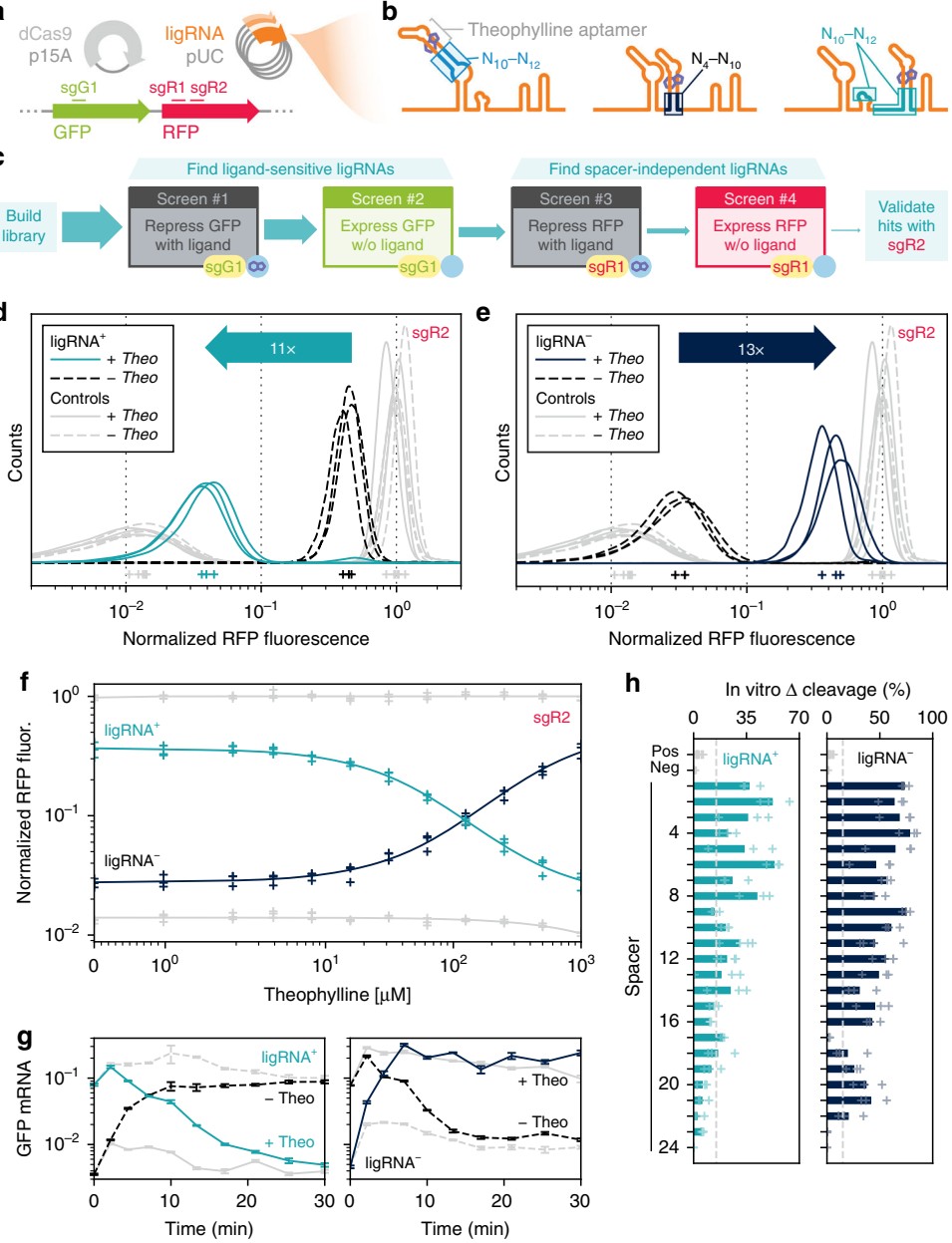

**Fig. 2** Identification of robust ligRNAs using CRISPRi-based gene repression in *E. coli*. **a** Components used in the CRISPRi assay. dCas9 and any ligRNAs were expressed from plasmids, while the fluorescent reporters (GFP and RFP) were chromosomally integrated. The DNA regions targeted by different spacers (sgG1, sgR1, sgR2) used to repress the fluorescent reporters are indicated. **b** Regions randomized in each ligRNA library. **c** Schematic of the screen used to isolate ligRNA+. sgG1, sgR1, and sgR2 refer to spacers targeting GFP and RFP, respectively (Supplementary Table 4). **d**, **e** Single-cell RFP fluorescence distributions for ligRNA+ (teal, **d**) and ligRNA− (navy, **e**) targeting RFP using the sgR2 spacer with (solid lines) and without (dashed lines) theophylline. Control distributions are in grey (positive control: optimized sgRNA scaffold;[51] negative control: G43C G44C[28]). The mode of each distribution is indicated with a plus sign. RFP fluorescence values for each cell are normalized by both GFP fluorescence for that cell and the modes of the un-repressed control populations (i.e. *apo* and *holo*) measured for that replicate. **f** Efficiency of CRISPRi repression with increasing theophylline concentrations for ligRNA+ (teal) and ligRNA− (navy). Controls are in grey. The fluorescence axis is the same as in **d** and **e**. The fits are to a two-state equilibrium model. **g** GFP mRNA levels after the addition or removal of theophylline. GFP mRNA levels were measured by qPCR and are normalized to 16S rRNA levels. Solid, colored lines: theophylline added at $t = 0$. Black, dashed lines: theophylline removed at $t = 0$. Grey lines: theophylline present (solid) or absent (dashed) for the whole experiment. Error bars reflect standard deviations from 3 technical replicates. **h** Change in the percentage of DNA cleaved in vitro in the presence and absence of theophylline for ligRNA+ and ligRNA− for 24 representative spacers. Bar heights represent the mean of three or four measurements with a single spacer (except for spacer #24, where $n = 2$, Supplementary Table 5). Data for each replicate are shown as faded plus marks. Pos and Neg denote the positive and negative control sgRNA scaffolds (Supplementary Table 4). The control bars combine data for all 24 spacers

removing it. Likewise, with ligRNA⁻, the response was 50% complete after 3.3 and 7.9 min, respectively. (For comparison, mRNAs have half-lives between <1 and 16 min in *E. coli*[33].) The rapid response times for the relief of repression were particularly surprising, since the dwell-time for dCas9 bound to DNA is estimated to be at least 45 min[34–36]. Possible explanations for the rapid response include RNA polymerase actively dislodging dCas9 from the DNA or the dCas9/ligRNA complex being affected by the mutations in the ligRNA. In either case, ligRNAs are a promising tool for applications requiring fast gene regulation.

**ligRNA function with different spacers**. Because RNA devices are known to be sensitive to sequence context[37], we tested ligRNA⁺ and ligRNA⁻ with 24 different spacers using the in vitro DNA cleavage assay (Fig. 2h, Supplementary Fig. 7, Supplementary Table 5). We found that both ligRNA⁺ and ligRNA⁻ respond to theophylline for the majority of the tested spacers (15 and 21 out of 24 spacers for ligRNA⁺ and ligRNA⁻, respectively). For the few spacers that did not function, we hypothesized that base-pairing of the spacer sequence with the aptamer might explain the lack of sensitivity to theophylline. To address this question, we predicted the affinity between each spacer and the aptamer (with its associated linker) for both ligRNAs using ViennaRNA[29] (Supplementary Fig. 8). For ligRNA⁺ constructs, the correlation between the duplex free energy prediction and theophylline

sensitivity was negligible. However, for ligRNA⁻ constructs increased predicted affinity of the spacer for the aptamer sequence correlated with a smaller change in Cas9-mediated DNA cleavage in response to theophylline. This analysis suggested that spacers with predicted affinity for the aptamer could interfere with switching of the ligRNA⁻ function, potentially limiting the space of sequences that could be targeted. Similar considerations also apply to the standard sgRNA scaffold, where internal pairing within the sgRNA sequence has been shown to affect CRISPR efficacy[38]. These considerations nonetheless provide useful design criteria for functional spacers. Taken together, these results suggest that ligRNAs should be capable of regulating most genes, especially those that can be targeted by multiple spacers.

**Regulation of endogenous genes in different bacteria**. We next tested the ability of the ligRNAs to regulate endogenous genes (rather than the chromosomally integrated fluorescent proteins used in our screen) in two different bacterial species. We first tested 4 different spacers targeting the endogenous *lac* operon (lacZ, lacI, A-site and P-site) using a β-galactosidase assay (Fig. 3a, Supplementary Fig. 9). ligRNA⁺ was functional at all loci, and ligRNA⁻ successfully targeted the two sites in lacZ and lacI (Fig. 3b). To determine whether ligRNAs also function in species other than *E. coli*, we created strains of *Pseudomonas aeruginosa* UCBPP-PA14 expressing both dCas9 and ligRNA⁺

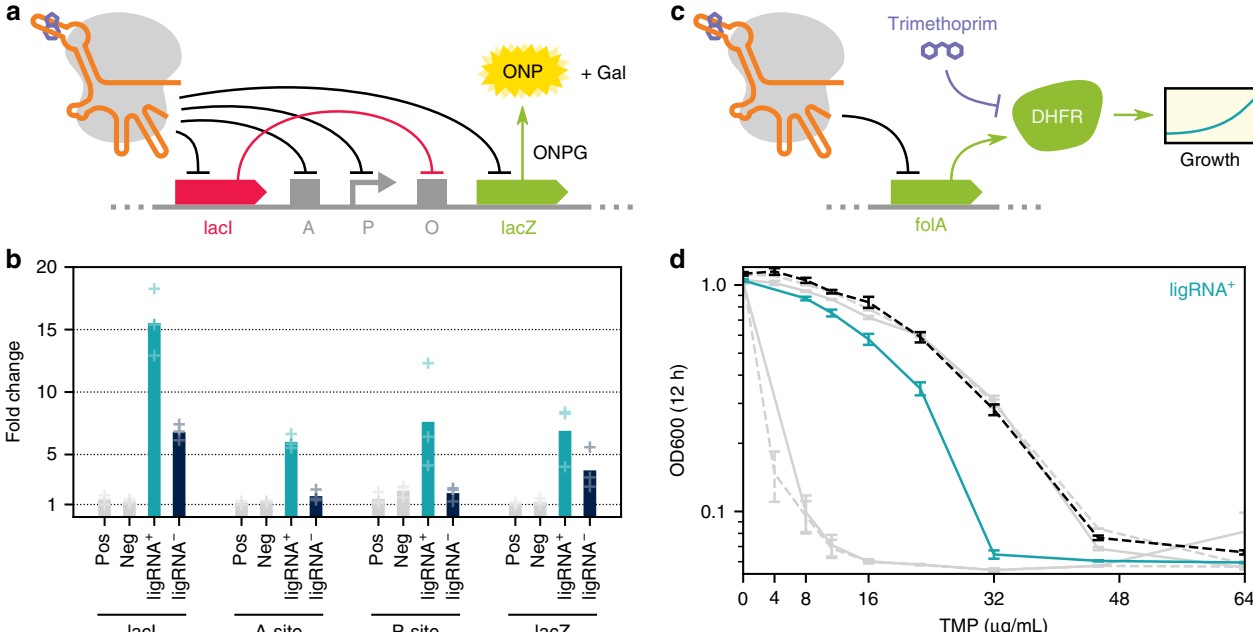

**Fig. 3** ligRNAs function in the context of the endogenous loci in different bacteria. **a** Schematic showing sites in the *E. coli lac* operon (the genes *lacI* and *lacZ*, the transcriptional operator sites A and O, and the promoter P driving LacZ expression) targeted by ligRNAs (black arrows). Note that LacI inhibits expression of LacZ (red arrow), so the ligRNAs targeting *lacI* are expected to increase LacZ expression when active (Supplementary Fig. 9). Expression of LacZ is detected by conversion of o-nitrophenol-β-D-galactopyranoside (ONPG) into o-nitrophenyl (ONP, yellow) and galactose (Gal). **b** Fold change in LacZ activity for *E. coli* grown with or without theophylline, expressing either a control sgRNA (pos or neg, grey), ligRNA⁺ (teal), or ligRNA⁻ (navy), with spacers targeting the indicated part of the *lac* operon. Spacer sequences were taken from Qi et al.[2] and are listed in Supplementary Table 4. Bar heights represent the mean of three biological replicates. Data for each replicate are shown as faded plus marks. The direction of the fold change (i.e. holo/apo or apo/holo) was chosen such that the resulting value is greater than one. ligRNA⁺ responds to theophylline for all 4 spacers, while ligRNA⁻ does so only for the two genes *lacZ* and *lacI*. **c** Schematic illustrating a growth assay in *P. aeruginosa* with ligRNA⁺ targeting the endogenous dihydrofolate reductase (DHFR) gene (*folA*). DHFR is neccesary for growth (green arrows) in the absence of thymidine, and is inhibited by the antibiotic trimethoprim (purple arrow). **d** Growth (OD600) of *P. aeruginosa* at 12 h in the presence (teal, solid) and absence (black, dashed) of theophylline at the indicated concentrations of trimethoprim (TMP). Inhibition of FolA by ligRNA⁺ in the presence of theophylline lowers the minimum concentration of trimethoprim needed to slow growth. Controls are in grey (high MIC: wildtype; low MIC: optimized sgRNA scaffold) and are also in the presence (solid) and absence (dashed) of theophylline. The spacer sequence is listed in Supplementary Table 4. Error bars are standard deviations of at least three biological replicates

targeting dihydrofolate reductase (DHFR). Repression of DHFR via CRISPRi lowers the minimal inhibitory concentration (MIC) of the antibiotic trimethoprim, which targets DHFR[39] (Fig. 3c). As expected, the ligRNA[+] strain consistently exhibited a lower MIC in the presence of theophylline (Fig. 3d).

**Independent control of multiple genes with multiple ligands**. A key advantage of regulating CRISPR-Cas9 using the sgRNA instead of the protein is the ability to independently control different genes with different ligands in the same Cas9 system. To test this idea, we replaced the theophylline (theo) aptamer in the ligRNAs with the 3-methylxanthine (3mx) aptamer (the resulting sgRNA constructs were termed ligRNA[±3mx]). While the theophylline aptamer is recognized by both ligands, the 3-methylxanthine aptamer is specific to its ligand[40]. Since the aptamers differ in only one position, the replacement of the theophylline aptamer with the 3-methylxanthine aptamer was straightforward and led to a 3-methylxanthine-sensitive ligRNA[−] variant without further optimization. (ligRNA[+] also remained functional with the 3-methylxanthine aptamer but exhibited an undesirable albeit small ~2-fold response to theophylline, Supplementary Fig. 10). We used the two ligRNA[−] variants to construct a system that expresses GFP upon addition of theophylline and expresses GFP and RFP when both ligands are added (Fig. 4a). We then performed a timecourse where we sequentially activated, deactivated, and reactivated both reporter genes using

ligRNAs and observed the expected temporal expression program (Fig. 4b). Going further, we also created thiamine-responsive ligRNAs by repeating our FACS screens (Fig. 2) with libraries containing the thiamine pyrophosphate aptamer (Supplementary Fig. 11, Supplementary Table 2, Supplementary Table 3). Although these ligRNAs have a narrower dynamic range (6-fold) than the theophylline or 3-methylxanthine ligRNAs, they demonstrate that our overall strategy for creating ligRNAs is applicable to other ligands. Taken together, these results suggest that concurrent control of multiple genes using ligRNAs responsive to different ligands is possible.

## Discussion
While ligRNAs function robustly in bacteria, transferring them to eukaryotic systems will require further optimization. There are two key requirements for CRISPR/Cas9 to be controllable using ligand-responsive sgRNAs: First, the system has to be limited by the concentration of active sgRNAs, and second, this concentration has to change in response to ligand to reach sufficient levels for activity only in the "on" state. In contrast, we find that in eukaryotic cells the ligRNAs are inactive whether or not theophylline is present (Supplementary Fig. 12, Supplementary Fig. 13). This observation is unlikely due to the inability of theophylline to cross cell membranes, or to other general factors that may interfere with aptamer function, as the theophylline aptamer has been used to regulate gene expression in mammalian cells in multiple different contexts[41]. Moreover, it is also unlikely that normal CRISPR-Cas9 activity is inhibited in eukaryotes under our conditions, as control sgRNAs targeting the same sites are functional in our mammalian cell and *S. cerevisiae* experiments (Supplementary Fig. 12, Supplementary Fig. 13). One possible explanation for the lack of transferability to eukaryotic systems is that the ligRNAs have decreased affinity for Cas9 (as both ligRNAs contain mutations in regions that are important for Cas9 binding, the nexus in particular) and therefore do not reach the required active state concentration in the "on" state in eukaryotes. Future work might use screens similar to those described in Fig. 2 to develop suitable ligRNAs for eukaryotic systems. Nevertheless, there are already many useful applications for ligRNAs in bacteria. For example, many species of bacteria do not have facile genetic controls available, and ligRNAs provide such controls with a minimal footprint. Moreover, temporally controlled gene expression programs are thought to be important for key biological processes in bacteria[42], and ligRNAs provide a way to conduct large-scale screens to probe these programs and their role in the interactions between bacteria and their environments[43].

In conclusion, ligRNAs provide control of both gene repression and gene activation and can be multiplexed for differential control of genes in the same system (Fig. 4). The study of subtle effects in complex biological systems will increasingly require the ability not just to probe individual genes, or to knock down different sets of genes, but to tune the expression of many different genes with fine temporal precision. ligRNAs provide this capability by adding ligand- and dose-dependent control of individual sgRNAs to the already powerful CRISPR-Cas9 technology.

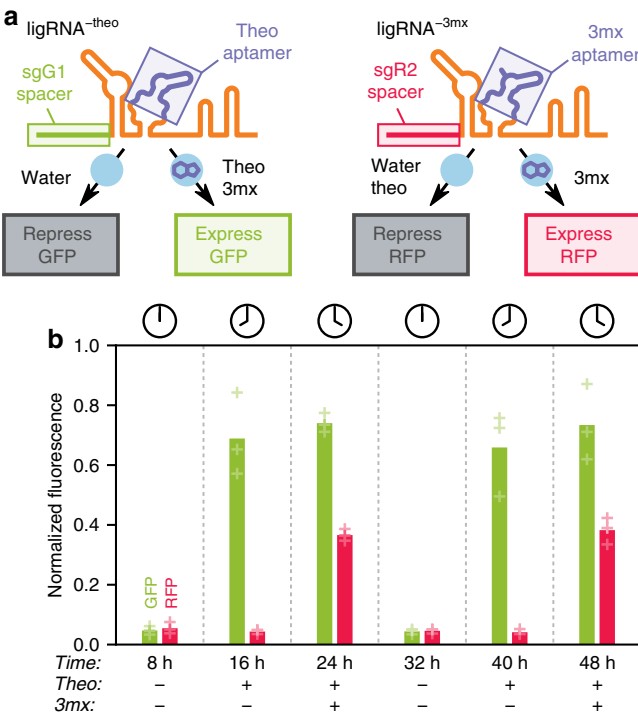

**Fig. 4** Multiplexed temporal control of two genes with two ligands. **a** Schematic illustrating the constructs and the expected consequences of adding theophylline (theo) and 3-methylxanthine (3mx) for each fluorescent reporter. **b** GFP and RFP fluorescence measured at indicated time points by flow cytometry. Presence of theophylline leads to GFP expression; addition of 3-methylxanthine separately at a different time point also triggers RFP expression. Both effects are reversible (GFP and RFP are repressed when both ligands are absent) and expression can be triggered a second time with ligand addition. Bar heights represent the mean of three biological replicates. Data for each replicate are shown as faded plus marks. Unlike in Fig. 2, fluorescence is normalized by side-scatter because both fluorescent channels are being manipulated

## Methods
**Constructs**. All experiments used Cas9 from *S. pyogenes* (called Cas9 throughout), with mutations D10A and H840A for CRISPRi experiments (dCas9). Sequences of relevant ligRNAs, aptamers, spacers and controls are listed in Supplementary Table 4.

**In vitro DNA cleavage assay**. Linear, double-stranded template DNA was acquired either by ordering gBlocks® Gene Fragments from IDT (experiments in

Fig. 1d, e) or by cloning the desired sequence into a pUC vector and digesting it with EcoRI and HindIII (experiments in Fig. 2h). Each construct contained a T7 promoter and a spacer that began with 3 5′ Gs (Supplementary Table 5) for efficient transcription by T7 polymerase. DNA template (10–50 ng) was transcribed using the HiScribe™ T7 High Yield RNA Synthesis Kit (NEB E2040S) and unincorporated ribonucleotides were removed with Zymo RNA Clean & Concentrator™-25 spin columns (Zymo R1018).

Target DNA was prepared using inverse PCR to clone the appropriate sequence into a modified pCR2.1 vector ~2.1 kb downstream of its XmnI site. The vector was then digested with XmnI (NEB R0194S) as follows: mix 43.5 μL ≈500 ng/μL miniprepped pCR2.1 DNA, 5.0 μL 10× CutSmart buffer, and 1.5 μL 20 U/μL XmnI; incubate at 37 °C until no uncleaved plasmid is detectable on a 1% agarose gel (usually 30–60 min); dilute to 30 nM in 10 mM Tris–Cl, pH 8.5; store at −20 °C.

For the Cas9 reaction, we adapted the following protocol from ref. [28]; mix 5.0 μL water or 30 mM theophylline (in water) and 1.5 μL 1.5 μM sgRNA (in water); incubate at 95 °C for 3 min, then at 4 °C for 1 h; prepare Cas9 master mix for 40 reactions: 241.0 μL water, 66.0 μL 10x Cas9 buffer (NEB B0386A), and 1.0 μL 20 μM Cas9 (NEB M0386T); add 7.0 μL Cas9 master mix; incubate at room temperature for 10 min; add 1.5 μL 30 nM target DNA; pipet to mix; incubate at 37 °C for 1 h; prepare quenching master mix: 4.68 μL 20 mg/mL RNase A (Sigma R6148), 4.68 μL 20 mg/mL Proteinase K (Denville CB3210-5), and 146.64 μL 6x Orange G loading dye via master mix; add 3 μL quenching master mix; incubate at 37 °C for 20 min, then at 55 °C for 20 min; run the entire reaction (18 μL) on a 1% agarose/TAE/GelRed gel at 4.5 V/cm for 70 min.

Band intensities were quantified using Fiji (1.51r)[44]. The background was subtracted from each image using a 50-pixel rolling ball radius. The fraction of DNA cleaved in each lane (f) was calculated as follows (pixels$_{2kb}$ and pixels$_{4kb}$ are the intensities of the cleaved and uncleaved bands, respectively):

$$f = \frac{\text{pixels}_{2kb}}{\text{pixels}_{4kb} + \text{pixels}_{2kb}} \quad (1)$$

The change in cleavage due to ligand (Δf) was calculated as follows ($f_{apo}$ and $f_{holo}$ are the fractions of DNA cleaved in the reactions without and with theophylline, respectively):

$$\Delta f = f_{theo} - f_{apo} \quad (2)$$

**CRISPR-Cas9-based repression (CRISPRi) assay in E. coli.** The strain used for all CRISPRi experiments was E. coli MG1655 with dCas9 (containing the D10A and H840A mutations) and ChlorR on a p15A plasmid (pgRNA-bacteria, Addgene 44251), sgRNA (pdCas9-bacteria, Addgene 44249), and sfGFP[45], mRFP[46], and KanR chromosomally integrated at the nsfA locus. This strain was originally described by Qi et al.[2].

Overnight cultures of the CRISPRi strain above were inoculated from freshly picked colonies in 1 mL Lysogeny Broth (LB) medium with 100 μg/mL carbenicillin (100 mg/mL stock in 50% EtOH) and 35 μg/mL chloramphenicol (35 mg/mL stock in EtOH). The next morning, fresh cultures were inoculated in 15 mL culture tubes or 24-well blocks by transferring 4 μL of overnight culture into 1 mL EZ Rich Defined Medium (Teknova M2105) with 0.1% glucose, 1 μg/mL anhydrotetracycline, 100 μg/ mL carbenicillin, 35 μg/mL chloramphenicol, with or without 1 mM theophylline (30 mM stock in water). These cultures were then grown for 8 h at 37 °C with shaking at 225 rpm before GFP (488 nm laser, 530/30 filter) and RFP (561 nm laser, 610/10 filter) fluorescence were measured using a BD LSRII flow cytometer. Approximately 10,000 events were recorded for each measurement. Biological replicates were performed on different days using different colonies from the same transformation.

Cell distributions were obtained by computing a Gaussian kernel density estimation (KDE) over the base-10 logarithms of the measured fluorescence values. The mode was considered to be the center of each distribution (e.g. for determining fold changes) and was obtained through the Broyden–Fletcher–Goldfarb–Shanno (BFGS) maximization of the KDE. Dose response curves were fit to the Hill equation (y is the normalized fluorescence, x in the theophylline concentration, EC50 is the inflection point, and $y_{min}$ and $y_{max}$ are the lower and upper asymptotes of the fit):

$$y = y_{min} + \frac{y_{max} - y_{min}}{1 + \text{EC50}/x} \quad (3)$$

The script used for data analysis is available from GitHub: https://raw.githubusercontent.com/kalekundert/ligrna/master/flow_cytometry/fold_change.py

**Identification of functional ligRNAs in E. coli.** To generate libraries, randomized regions were inserted into the sgRNA using inverse polymerase chain reaction (PCR) with phosphate-modified and high-performance liquid chromatography (HPLC)-purified primers containing degenerate nucleotides.

Electrocompetent cells were prepared as follows: make "low-salt" Super Optimal Broth (SOB) medium: 20 g bacto-tryptone, 5 g bacto-yeast extract, 2 mL 5 M NaCl, 833.3 μL 3 M KCl, water to 1 L, pH to 7.0 with NaOH, autoclave 30 min at 121 °C; pick a fresh colony and grow overnight in 1 mL SOB; in the morning, inoculate 1 L SOB with the entire overnight culture; grow at 37 °C with shaking at 225 rpm until OD = 0.4 (≈4 h); place cells in an ice bath for 10 min; wash with 400 mL pre-chilled

water, then 200 mL pre-chilled water, then 200 mL pre-chilled 10% glycerol; resuspend in a total volume of 6 mL pre-chilled 10% glycerol; make 100 μL aliquots; flash-freeze and store at −80 °C. Electrocompetent cells were transformed as follows: thaw competent cells on ice for 10 min; pipet once to mix cells with 2 μL ≈250 ng/μL library plasmid; shock at 1.8 kV with a 5 ms decay time; immediately add 1 mL pre-warmed SOB with catabolite repression (SOC) medium; recover at 37 °C for 1 h; dilute into selective liquid media and grow at 37 °C with shaking at 225 rpm overnight. After PCR and ligation, libraries were transformed into electrocompetent Top10 cells, mira-prepped[47], sequenced, combined to achieve approximately equal representation of variants based on library size and DNA concentration, and transformed into electrocompetent MG1655 cells already harboring the dCas9 plasmid.

Cells were grown as for the CRISPRi assay, but when starting new cultures, care was taken to subculture at least 10× more cells than the size of the library (often 200 μL). Sorting was done using a BD FACSAria II cell sorter. Sorting was no slower than 1000 evt/s and no faster than 20,000 evt/s, with the slower speeds being more accurate and the faster speeds being necessary to sort large libraries. Gates were drawn based on the position of the control population if possible, and based on the most extreme library members otherwise. Typically the gates included between 1% and 5% of the population being sorted. All gates were drawn diagonally in GFP vs. RFP space. Sorted cells were collected in 1 mL SOC at room temperature and, after sorting, were diluted into selective media and grown at 37 °C with shaking at 225 rpm overnight.

Our screening protocol for ligRNA⁺ was as follows: Pool libraries 23–28 from Supplementary Table 2. First screen: grow without ligand, gate for GFP expression, sort 10,000 evt/s for 3.5 h. Second screen: grow with ligand, gate for GFP repression, sort 1500 evt/s for 70 min. Third screen: grow without ligand, gate for GFP expression, sort 1700 evt/s for 10 min. Fourth screen: grow with ligand, gate for GFP repression, sort 1000 evt/s for 2 min. Fifth screen: grow without ligand, gate for GFP expression, sort 5000 evt. Plate cells and test 96 individual colonies using the CRISPRi assay. Miniprep and sequence the 20 selected designs with the largest response to theophylline. Only one unique sequence was identified, and it did not function with the sgR1 spacer (Supplementary Table 3). Note that we did not change the spacer in between the second and third screens for this library. We then designed libraries 29–30 (Supplementary Table 2) to keep the stem identified in the previous screen of libraries 23–28 and to randomize other regions of the sgRNA that might be participating in ligand-dependent base-pairing. First screen: grow with ligand, gate for GFP repression, sort 4000 evt/s for 2 h. Second screen: grow without ligand, gate for GFP expression, sort 1500 evt/s for 1 h. Change the spacer from sgG1 to sgR1 (Supplementary Table 4) using inverse PCR. Third screen: grow with ligand, gate for RFP repression, sort 2000 evt/s for 10 min. Fourth screen: grow without ligand, gate for RFP repression, sort 10,000 evt. Plate cells and test 96 individual colonies using the CRISPRi assay. Miniprep and sequence the 15 selected designs with the largest response to theophylline, then test those designs with four different spacers (sgG1, sgR1, sgG2, sgR2). There was only 1 duplicate sequence, and 8 of the sequences had acquired unexpected mutations outside of the randomized region. The majority of these hits were tested with four different spacers (sgG1, sgR1, sgG2, and sgR2). ligRNA⁺ and ligRNA⁻ performed best (none of the hits performed well with the sgG2 spacer.) Details on all library hits and validation with different spacers can be found in Supplementary Table 3.

Our screening protocol for ligRNA⁻ was as follows: Pool libraries 7–22 from Supplementary Table 2. First screen: grow without ligand, gate for GFP repression, sort 18,000 evt/s for 1 h. Second screen: grow with ligand, gate for GFP expression, sort 1000 evt/s for 10 min. Third screen: grow without ligand, gate for GFP repression, sort 1000 evt/s for 10 min. Fourth screen: grow with ligand, gate for GRP expression, sort 1000 evt/s for 10 min. Fifth screen: grow without ligand, gate for GFP repression, sort 1500 evt/s for 7 min. Plate cells and test 96 individual colonies using the CRISPRi assay described above. Miniprep and sequence the 20 selected designs with the largest fold response to theophylline. In this group there were only 9 unique sequences. ligRNA⁻ appeared 5 times and had the largest response to theophylline (Supplementary Table 3). Note ligRNA⁻ is functional with other spacers (Fig. 2h) despite the fact that we did not change the spacer in between the second and third screens for this library.

Thiamine-dependent ligRNAs were identified using the same screening strategy described above. The library sequences are given in Supplementary Table 2 and use the thiamine pyrophosphate aptamer described by Wieland et al.[48]. The growth medium was prepared as follows: 1× MOPS (Teknova M2101), 1× K₂HPO₄ (Teknova M2102), 1× Supplement EZ (Teknova M2104), 0.4% glucose (Teknova G0520), 0.2% casamino acids, 1 μg/mL anhydrotetracycline, 100 μg/mL carbenicillin, 35 μg/mL chloramphenicol, 500 μM thiamine dissolved in water (holo medium only). Note that thiamine is metabolized to thiamine pyrophosphate.

**mRNA timecourse after addition and removal of theophylline.** Overnight and day cultures were setup as described for the flow cytometry experiments, except that 88 μL of overnight culture were used to inoculate 22 mL of each day culture for this experiment. The day cultures were grown at 37 °C for 4 h30 in a 125 mL flask with shaking at 225 rpm. After the growth period, timepoints were processed as follows, while making an effort to keep the cells at 37 °C (i.e. in a 37 °C heat block) as much as possible: split the day culture into 19 1 mL aliquots (1 for the 0:00

timepoint, and 2 for each other timepoints); pellet the aliquots at 16,000 g for 30 s; remove the supernatant; simultaneously resuspend the 2 pellets for each timepoint (except the 0:00 timepoint) in medium with and without theophylline; 1:05 before each timepoint: load the resuspended cells corresponding to that timepoint into a tabletop centrifuge, spin at 16,000 g for 15 s, remove the supernatant, and add 1 mL TRIzol (Invitrogen 15596026). Each timepoint was considered to be the time at which the TRIzol was added. In some experiments, the lysed cells were stored in TRIzol at 4 °C overnight. Only a single biological replicate was performed.

Total cellular RNA was extracted using TRIzol and concentrated by isopropanol precipitation with GlycoBlue (Invitrogen AM9516), following the manufacturer's instructions for both steps and resuspending in a final volume of 10 μL nuclease-free water. RNA concentration and purity were determined by spectrophotometric analysis (Thermo ND-ONE-W). Typical yield: 483.29 ng/μL (25th percentile) to 945.65 ng/μL (75th percentile); A260/A280: 1.86 ± 0.06 (standard deviation). Reverse transcription (RT) was performed as follows: mix 1 μg RNA, 0.5 μL ezDNase (Invitrogen 11766051), 0.5 μL 10× ezDNase buffer, nuclease-free water to 5 μL; incubate at 37 °C for 2 min; centrifuge; place on ice; add 2.0 μL SuperScript IV VILO master mix (Invitrogen 11766050) and 3.0 μL nuclease-free water; incubate at 25 °C for 10 min, 50 °C for 15 min, and 85 °C for 5 min; dilute to 100 μL with nuclease-free water. In some experiments, the reverse-transcribed DNA was stored overnight at 4 °C. The RT reaction was primed with random nucleotide hexamers present in the master mix.

Quantitative real-time polymerase chain reaction (qPCR) reactions were setup manually in 384-well plates with white wells (Biorad HSP3805) as follows: 8.0 μL water, 3.0 μL cDNA from the diluted RT reaction (concentration not determined), 0.75 μL 10 μM forward primer, 0.75 μL 10 μM reverse primer, 12.5 μL 2× Power SYBR Green PCR master mix (Applied Biosystems 4367659). Six reactions were prepared for each sample: 3 technical replicates with primers targeting sfGFP, and 3 technical replicates with primers targeting the reference gene (16 S rRNA). Plates were heat-sealed (Biorad 1814030). Amplification was measured using a CFX384 Real-Time PCR Detection System (Biorad) with the following temperature schedule: 95 °C for 10 min, 40 cycles of 95 °C for 15 s and 60 °C for 1 min. Melting curves were measured from 65 °C to 95 °C in steps of 0.5 °C.

Primers (Supplementary Table 6) were ordered from Elim Biopharm with no modifications. 8 annealing temperatures (55.7–65.5 °C) and 3 primer concentrations (50 nM, 300 nM, 900 nM) were tested. Optimal amplification (i.e. lowest $C_q$) was observed with annealing at 60.0 °C and primers at 300 nM. The efficiency for each pair of PCR primers is determined by performing qPCR on at least 5 10-fold serial dilutions of reverse-transcribed total cellular RNA (Supplementary Fig. 14).

For comparison with the controls described below, all 228 experimental samples had $C_q$ values between 10.10–16.48 (sfGFP primers) and 7.47–9.47 (16S rRNA primers). To test for contamination of the PCR reagents, we performed "no-template controls" (NTC) in triplicate on both plates in the experiment. 6 of 6 reactions with the sfGFP primers and 2 of 6 reactions with the 16S rRNA primers showed no amplification after 40 cycles. The remaining 4 reactions had an average $C_q$ value of 38.75 ± 0.64 (all uncertainties are standard deviations), negligible compared to the control samples. To test for genomic DNA contamination, we performed "no reverse transcriptase" (NRT) controls in triplicate on both plates. These reactions had an average $C_q$ values of 29.65 ± 2.37 (sfGFP primers) and 26.84 ± 3.33 (16S rRNA primers), negligible compared to the experimental samples. To test for the specificity of our sfGFP primers, we performed qPCR on cDNA extracted from a strain identical to the one used in our experiments, but missing the sfGFP gene. We performed this control in triplicate on both plates. These reactions had average $C_q$ values of 26.15 ± 2.76 (sfGFP primers) and 8.77 ± 0.26 (16S rRNA primers). As expected, the sfGFP values are negligible compared to the experimental samples, while the 16S rRNA values are in line with the experimental samples.

$C_q$ values were determined by the CFX Maestro software (Biorad, version 4.1) using a single threshold value. GFP mRNA levels were normalized relative to the reference gene (16S rRNA) as follows ($\mu[C_q,X]$ is the average $C_q$ value for the technical replicates of gene X; $\sigma[C_q,X]$ is the standard deviation for the technical replicates of gene X; GFP mRNA is the value reported in Fig. 2g):[49]

$$\mu\left[\Delta C_q\right] = \mu\left[C_q, \text{GFP}\right] - \mu\left[C_q, 16\text{S}\right] \tag{4}$$

$$\sigma\left[\Delta C_q\right] = \sqrt{\sigma\left[C_q, \text{GFP}\right]^2 + \sigma\left[C_q, 16\text{S}\right]^2} \tag{5}$$

$$\text{GFP mRNA} = 2^{-\mu\left[\Delta C_q\right] \pm \sigma\left[\Delta C_q\right]} \tag{6}$$

The script used for data analysis is available from GitHub:
https://raw.githubusercontent.com/kalekundert/ligrna/notebook/master/20181115_measure_ligrna_induction_time_scale/gfp_mrna_vs_time.py

**Quantification of sgRNA expression levels.** Overnight and day cultures were setup as described for the flow cytometry experiments, except that 100 μL of overnight culture was used to inoculate 5 mL of day culture. The day cultures were grown for 2 h at 37 °C with shaking at 225 rpm. Total cellular RNA was extracted and prepared for qPCR as described above.

15 pairs of qPCR primers, comprising 3 forward primers and 5 reverse primers, were designed manually (Supplementary Table 6). The same 3 forward primers were used for all of the sgRNA templates. 2 of the reverse primers were used for the control sgRNAs and ligRNA⁻, while the other 3 were used for ligRNA⁺. This was necessary due to the insertion of the aptamer in the 3′ region of ligRNA⁺. Efficiencies were measured (as described above) for the primer pairs that gave the best amplification for each target (Supplementary Fig. 14).

qPCR was performed as described above, except that no ezDNase was added to the RT reaction and a QuantStudio3 qPCR machine (Applied Biosystems A28137) was used.

Data analysis was performed as described above. The script used for analysis is available from GitHub:
https://raw.githubusercontent.com/kalekundert/ligrna/master/notebook/20180925_quantify_sgrna_levels/20181002_sgrna_qpcr.py

**Test of different spacers in vitro.** The spacers for this assay were chosen by a script that generated uniformly random sequences, scored them using a previously published machine learning approach for designing functional sgRNAs[50], and kept only those that scored higher than 0.5 (the median). This approach was designed to produce spacers that were as unbiased as possible, while still being likely to function as expected in the positive (optimized sgRNA scaffold[51]) and negative (G43C G44C[28]) controls shown in Supplementary Table 4. Note that only spacers used in the in vitro cleavage assay were generated randomly. All other spacers were designed to target the respective genes (gfp, rfp, folA, lacI, lacZ) or genomic target sites as indicated. The average cleavage for the positive controls was 93%, and the lowest cleavage for any of the positive controls was 78% (Supplementary Table 5).

The script used to design spacers is available on GitHub:
https://raw.githubusercontent.com/kalekundert/ligrna/master/notebook/20170329_test_multiple_spacers/doench16/pick_doench16_spacers.py

**RNA secondary structure predictions.** Secondary structure predictions were performed using the RNAfold program from the ViennaRNA package (version 2.4.3). The structures reported here are minimum free energy (MFE) predictions, although centroid and maximum expected accuracy (MEA) structures from partition function calculations were nearly identical in every case. The holo state was simulated using soft constraints: a -9.21 kcal/mol bonus was granted for forming the base pair flanking the aptamer. This bonus corresponds to the 320 nM affinity of the theophylline aptamer for its ligand[30].

The command-lines used for the apo and holo states, respectively, are given below:

```
$ RNAfold --partfunc --MEA
$ RNAfold --partfunc --MEA --motif \
"GAUACCAGCCGAAAGGCCCUUGGCAGC,(…(((((((….)))…)))…),-9.212741321099747"
```

Free energy predictions for Supplementary Fig. 3 were performed using the RNAduplex program from the ViennaRNA package (version 2.4.3):

```
$ RNAduplex
```

**CRISPRi assays targeting the E. coli lac operon.** Overnight cultures were setup as described for the flow cytometry experiments. The next morning, fresh cultures were inoculated in 24-well blocks by transferring 4 μL of overnight culture into 1 mL LB medium with 1 μg/mL anhydrotetracycline, 100 μg/mL carbenicillin, 35 μg/mL chloramphenicol, 0 or 1 mM theophylline, and 0 or 1 mM IPTG. Note that this medium contains no glucose, to avoid repression of the lac operon. These cultures were then grown for 6 h at 37 °C with shaking at 225 rpm. β-galactosidase activity was then measured as follows: spin the cells at 3500 g for 10 min; resuspend in 1 mL phosphate-buffered saline (PBS; 137 mM NaCl, 2.7 mM KCl, 10 mM Na₂HPO₄, 1.8 mM KH₂PO₄, pH 7.4); transfer 100 μL to a black-well, clear-bottom plate; adjust each well to OD600 = 0.15 by adding cells or PBS (then removing an equal volume to remain at 100 μL); add 100 μL Y-PER (Thermo 75768); mix well; incubate at room temperature for 1 h; add 100 μL β-galactosidase assay buffer (Thermo 75768); mix well; pop any bubbles using a hot needle; measure absorbance at 420 nm (A420) every minute for 45 min via plate reader (BioTek Synergy H1).

The A420 vs time data were fit using a linear regression to determine the initial rate of each reaction. Data points that appeared substrate-limited (often the case when A420 >1.2 before baseline correction) were removed from the fit (Supplementary Fig. 9). The slopes of these fits were used to calculate β-galactosidase activity in Miller units as follows ($A_{\text{Miller}}$ is activity in Miller units; $m$ is the slope of the aforementioned regression; $\text{OD}_{600}$ is the density of the culture; $V$ is the volume of the culture in mL):

$$A_{\text{Miller}} = \frac{1000 \times m}{\text{OD}_{600} \times V} \tag{7}$$

The script used to calculate activity is available on GitHub: https://raw.githubusercontent.com/kalekundert/ligrna/master/notebook/20180621_target_endogenous_loci/miller_units.py

**CRISPRi assays in *P. aeruginosa***. Strains of UCBPP-PA14 expressing dCas9 with positive control sgRNA or ligRNA+ targeting *folA* (Supplementary Table 4) were prepared as described by Peters et al.[39], except that the constitutive promoters J23117 (positive control) and J23115 (ligRNAs and negative control) from the Anderson promoter collection (http://parts.igem.org/Promoters/Catalog/Anderson) were used to drive dCas9 expression. The sequence of the positive control sgRNA was: CGCGCGGTTCTCGCCAAGGGGTTTAAGAGCTATGCTG GAAACAGCATAGCAAGTTTAAATAAGGCTAGTCCGTTATCAACTTGAAA AAGTGGCACCGAGTCGGTGCTTTTT (underlined: target sequence). Note that, compared to the ligRNAs in this experiment, the positive control expressed dCas9 with a weaker promoter and targeted a less-repressing region of *folA*. We could not create a positive control strain without these differences, presumably due to the toxicity of constitutive *folA* repression. The negative control for the PA14 experiments had no sgRNA, since the G63C,G64C mutant used in other experiments was not fully inactive in PA14.

To determine the minimum inhibitory concentration (MIC), overnight cultures of the PA14 strains were inoculated from freshly picked colonies in 1 mL LB medium with no antibiotics. The next morning, fresh cultures were inoculated in 15 mL culture tubes by transferring 40 μL of overnight culture into 2 mL Mueller–Hinton (MH) medium (Difco 275730). These cultures were grown for 2 h at 37 °C with shaking at 225 rpm, then growth in the presence of trimethoprim and theophylline was measured as follows: in the wells of a clear-flat-bottom 96-well plate, mix 50 μL 4× theophylline (where 1× ranges from 0 to 1 mM), 50 μL 4× trimethoprim (where 1× ranges from 0 to 64 μg/mL), and 100 μL of the above cultures diluted to OD600 = 0.005; cover each well with 50 μL mineral oil; spin at 3500 g for 1 min; incubate in a plate reader at 37 °C for 24 h with continuous shaking while measuring OD600 every 5 min.

**CRISPRi assay in *S. cerevisiae***. All yeast strains were constructed using the MoClo golden gate cloning framework and the Yeast Toolkit from[52]. The background strain was WCD230 (derived from BY4741 (MATa his3Δ1 leu2Δ0 met15Δ0 ura3Δ0)) with a larger fraction of the his3 gene removed[53]. All yeast strains contained the dCas9-Mxi1 inhibitor, fluorescent reporters, and ligRNA or control sgRNA construct, respectively (Supplementary Fig. 13a). The inhibitor cassette was integrated into the Ura3 locus and contained pGal1-dCas9-Mxi1 and pRnr2-GEM constructs that allowed for estradiol-inducible expression of dCas9-Mxi1[54]. The Mxi1 inhibition domain was derived from Addgene catalog number 46921[3] with synonymous mutations made to the E68 and T69 codons to render the construct compatible with golden gate cloning. The fluorescent reporter cassette was integrated into the His3 locus and contained pCcw12-sfGFP and pTdh3-mRFP. The guide cassette was integrated into the Leu2 locus and contained (tRNAPhe)-HDV Ribozyme-sgRNA. The HDV Ribozyme cleaved the ligRNA or control sgRNA from the rest of the transcript to prevent unwanted interactions[55]. The sgRNA was either positive or negative control sgRNA, ligRNA+, ligRNA+2, or ligRNA−.

Strains were plated from freezer stocks on SD-HIS (Yeast Nitrogen Base (YNB), Complete Synthetic Media (CSM) lacking histidine, 2% glucose) because the fluorescent reporter cassette integrated into the His3 locus caused a slight growth defect that was also present for an empty vector integrated in the same locus. After incubating at 30 °C, single colonies were grown overnight in 0.5 ml YPD (yeast extract, peptone, 2% glucose) in 96 well plates with 2 ml/well maximum capacity, shaking at 900 RPM at 30 C in an Infors HT Multitron Pro shaker. Saturated cultures were diluted 1:100 into 1 ml SDC (YNB, CSM, 2% glucose) in a new 2 ml 96 well plate and placed at 30 °C shaking at 900RPM for 2 h. Cells were then diluted 1:4 into 400 μl SDC with estradiol and either theophylline from a 30 mM stock or water in a new 2 ml 96 well plate. The final concentration of estradiol was 125 nM and the final concentration of theophylline when present was 2.5 mM. After 8 h shaking at 900RPM and 30 °C, cells were diluted in 1×TE buffer and analyzed on a flow cytometer (BD LSR II).

**Gene editing assay in mammalian cells**. HEK293T (293FT; Thermo Fisher Scientific) cells, and derived cell lines, were grown in Dulbecco's Modified Eagle Medium (DMEM; Corning Cellgro, #10-013-CV) supplemented with 10% fetal bovine serum (FBS; Seradigm #1500-500), and 100Units/ml penicillin and 100 μg/ml streptomycin (Pen-Strep; Life Technologies Gibco, #15140-122) at 37 °C with 5% $CO_2$. HEK293T and HEK-RT1 cells were tested for absence of mycoplasma contamination (UC Berkeley Cell Culture facility) by fluorescence microscopy of methanol fixed and Hoechst 33258 (Polysciences #09460) stained samples.

HEK293T-based genome editing reporter cells, referred to as HEK-RT1, were established in a two-step procedure[56,57]. In the first step, puromycin resistant monoclonal HEK-RT3-4 reporter cells were generated. In brief, HEK293T human embryonic kidney cells were transduced at low-copy with the amphotropic pseudotyped retrovirus RT3GEPIR-sh.Ren.713[58], comprising an all-in-one Tet-On system enabling doxycycline-controlled EGFP expression. After puromycin (2.0 μg/ml) selection of transduced HEK239Ts, 36 clones were isolated and individually assessed for (i) growth characteristics, (ii) homogeneous morphology, (iii) sharp fluorescence peaks of doxycycline (1 μg/ml) inducible EGFP expression, (iv) relatively low fluorescence intensity to favor clones with single-copy reporter integration, and (v) high transfectability. HEK-RT3-4 cells are derived from the clone that performed best in these tests. In the second step, HEK-RT1 cells were derived by transient transfection of HEK-RT3-4 cells with vectors encoding Cas9

and sgRNAs targeting puromycin, followed by identification of monoclonal reporter cell lines that are puromycin sensitive.

A lentiviral vector, referred to as pCF204, expressing a U6 driven sgRNA and an EFS driven Cas9-P2A-Puro cassette was based on the lenti-CRISPR-V2 plasmid[59], by replacing the sgRNA with an enhanced *Streptococcus pyogenes* Cas9 sgRNA scaffold[5]. All sgRNAs (sgRen71: TAGGAATTATAATGCTTATC, sgGFP1: CCTCGAACTTCACCTCGGCG, sgGFP9: CCGGCAAGCTGCCCGTGCCC) were designed with a G preceding the 20-nt guide for better expression, and cloned into the lentiviral vector using the BsmBI restriction sites. The lentiviral vectors expressing ligRNA+, ligRNA2+ and ligRNA− (referred to as pCF441, pCF442 and pCF443, respectively) were all based on pCF204, by replacing the SpyCas9 sgRNA scaffold with the respective ligRNAs using custom oligonucleotides (IDT), gBlocks (IDT), standard cloning methods, and Gibson assembly techniques. Lentiviral particles were produced using HEK293T packaging cells; viral supernatants were filtered (0.45 μm) and added to target cells. Transduced HEK-RT1 target cells were selected on puromycin (1.0 μg/ml).

GFP expression in HEK-RT1 reporter cells was induced using doxycycline (1 μg/ml; Sigma-Aldrich). Percentages of GFP-positive cells were assessed by flow cytometry (Attune NxT, Thermo Fisher Scientific), routinely acquiring 10,000-30,000 events per sample. Theophylline (Sigma-Aldrich, #T1633-50G) was used at the indicated concentrations, ranging from 0.1 mM to 10 mM. Note, theophylline concentrations of 5 mM and 10 mM resulted in considerable cellular toxicity in the HEK293T-based reporter cell line.

**Reporting summary**. Further information on research design is available in the Nature Research Reporting Summary linked to this article.

## Data availability
All relevant data are reported in the main text or Supplementary Information. The source data underlying Fig. 1d, e are provided as a Source Data file. Any additional data relevant to this manuscript are available from the authors upon reasonable request.

## Code availability
Scripts used in data analysis are available from Github:
https://raw.githubusercontent.com/kalekundert/ligrna/master/flow_cytometry/fold_change.py
https://raw.githubusercontent.com/kalekundert/ligrna/notebook/master/20181115_measure_ligrna_induction_time_scale/gfp_mrna_vs_time.py
https://raw.githubusercontent.com/kalekundert/ligrna/master/notebook/20180925_quantify_sgrna_levels/20181002_sgrna_qpcr.py
https://raw.githubusercontent.com/kalekundert/ligrna/master/notebook/20170329_test_multiple_spacers/doench16/pick_doench16_spacers.py
https://raw.githubusercontent.com/kalekundert/ligrna/master/notebook/20180621_target_endogenous_loci/miller_units.py

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

## Acknowledgements

We thank Anna Simon for discussing research that in part inspired the project, Kianna Zucker for generating constructs used in Fig. 2h, and Kyle Barlow and Anum Glasgow for experimental help. We would also like to thank Carol Gross and Jason Peters for discussions on use of CRISPR-Cas9 in bacteria, Andy May for discussions on sgRNA structure-function relationships, and Pedro Batista for discussions on quantifying RNA expression. The work was supported by National Institutes of Health (NIH) grants R01-GM110089 and R21-EB021453 to T.K. and a Medical Research Award to T.K. by the W.M. Keck Foundation. J.E.L. was supported by a Graduate Fellowship from the National Science Foundation (NSF GRFP). O.S.R., H.E.S. and T.K. are Chan Zuckerberg Biohub Investigators. C.F. is supported by a US National Institutes of Health K99/R00 Pathway to Independence Award (K99GM118909) from the National Institute of General Medical Sciences (NIGMS). B.M.H. is supported by the Post 9/11 GI Bill. J.A.D. is a Paul Allen Frontiers Group Distinguished Investigator and an HHMI Investigator.

## Author contributions

K.K. conceived the idea for the project, K.K. and T.K. conceived the experimental approach, K.K. designed and performed the majority of the experiments and analyzed the data with advice and contributions from JEL (CRISPRi multiplexing with different ligands), K.E.W. (ligRNA⁻ mechanism), B.L.O. (FACS selections), C.M.F. (in vitro DNA cleavage and qPCR experiments), and J.Q. and N.P. (*P. aeruginosa* experiments). CF carried out the mammalian cell line experiments. AHN and BMH carried out the yeast experiments. O.S.R., D.F.S., H.E.S., J.A.D. and T.K. provided advice, mentorship and resources. K.K. and T.K. wrote the manuscript with input from all authors.

## Additional information

**Competing interests:** C.F. is a co-founder of Mirimus, Inc. B.L.O. is a co-founder and employee of Scribe Therapeutics. D.F.S. is a co-founder of Scribe Therapeutics and a scientific advisory board member of Scribe Therapeutics and Mammoth Biosciences. J.A.D. is a co-founder of Caribou Biosciences, Editas Medicine, Intellia Therapeutics, Scribe Therapeutics, and Mammoth Biosciences. J.A.D. is a scientific advisory board

member of Caribous Biosciences, Intellia Therapeutics, eFFECTOR Therapeutics, Scribe Therapeutics, Synthego, Metagenomi, Mammoth Biosciences, and Inari. J.A.D. is a member of the board of directors at Driver and Johnson & Johnson. The remaining authors declare no competing interests.

