## [Peer Review File · Nature Communications]

Reviewers' Comments:

Reviewer #1:

Remarks to the Author:

Kundert et al. have developed a novel method to control *S. pyogenes* Cas9 activity via the use of small-molecule-responsive RNA aptamers. The authors use this platform to control Cas9 nuclease activity in vitro and to control Cas9-mediated repression at synthetic loci (GFP/RFP) in *E. coli*. The authors also show that this platform enables multiplexed control over repression of synthetic loci in *E. coli*. Overall, the technological platform is innovative and further it could have broad impact across multiple scientific disciplines. The work is also scientifically sound throughout. However, there are several opportunities to increase the impact of this work which would make the manuscript suitable for publication in *Nature Communications*. There are three general issues i) How generalizable is the technology to other spacers/species of Cas9 ii) Does the technology work at native loci in *E. coli*, and in other bacteria, iii) Why does the system fail in eukaryotic/mammalian cells. I have provided specific comments in connection to these issues below.

1. There appear to be relatively few spacer sequences tested in this manuscript, making it difficult to assess how this system would function across different targets. Unless I am mistaken, the total number of spacers tested in the main text is 28, broken down as follows:

Figure 1: 1 spacer in vitro (cutting) for aptamer designs.

Figure 2: 3 spacers in *E. coli* (dCas9 repression) for design of Lig+ and Lig- designs

Figure 2: 24 spacers in vitro (cutting) for Lig+ and Lig- designs

Figure 3: 2 but Same spacers for (dCas9) repression In *E. coli*.

This seems like too few to generalize the programmability of the system. Targeting 1 or 2 endogenous loci in *E. coli* would not only help to address this, but would also increase impact via the nuclease-targeting of native, endogenous loci.

2. Related to point 1, the following issues may decrease the applicability of this platform. Can the authors please comment on each to clarify.

A. For in vitro cleavage assays, all gRNA spacers begin with 3 Gs for "facile transcription by T7 polymerase".

This seems to bottleneck the gRNA search space from the outset of the study?

B. The author's state that "for ligRNA- constructs increased predicted affinity of the spacer for the aptamer sequence correlated with a smaller change in Cas9-mediated DNA cleavage in response to theophylline...This analysis suggested that spacers with predicted affinity for the aptamer could interfere with switching of the ligRNA- function, providing a useful design criterion for functional spacers."

Doesn't this phenomenon also limit the design space for the protospacers that would be compatible with this system?

C. The authors also comment that "spacers for this assay were chosen by a script that generated uniformly random sequences". Is this for all spacers used in this manuscript or only a certain subset?

3. Showing that the platform is extensible the TRACR portions of gRNAs from other bacterial species would substantially increase the impact of this manuscript and utility of this platform.

4. Especially given the importance that the authors place on being able to use this system in different bacteria. See "many species of bacteria do not have facile genetic controls available, and ligRNAs provide such controls with a minimal footprint. Moreover, temporally controlled gene expression programs are thought to be important for key biological processes in bacteria and ligRNAs provide a way to conduct large-scale screens..."

5. The platform does not work in eukaryotes/mammalian cells. This is a major limitation and is

counter to many of the authors statements regarding the potential for broad applicability of this technology. For example "We envision that this method will be broadly useful for regulating essentially all applications of CRISPR-Cas9-mediated biological engineering". I think that tempering some of that language is appropriate.

6. Related to point #5 – two items in particular would be of tremendous value to the field and would strengthen this manuscript substantially. In order of importance:

A. Getting the system to work in mammalian cells (or even yeast) is critical.

B. Understanding why the system does not work in eukaryotes/mammalian cells would also be very useful to inform future studies.

The authors suggest that "While ligRNAs function robustly in bacteria, transferring them to eukaryotic systems will require further optimization which could be achieved using methods similar to our optimization (Figure 2) of the initial rational designs (Figure 1).4

I think that this would be a very important way to augment this work and potentially clarify the next steps in transferring this technology to eukaryotes. Especially since the authors are already well-versed in the assays/techniques in-house.

Some relevant and important items:

Are all the necessary components for this system appropriately expressed in eukaryotes?

Does chromatin affect activity?

Does the in vitro assay work with mammalian targets in mammalian cell lysates (i.e. is something inhibitory to efficacy in mammalian cells)?

7. (9 out of 24) 37.5% of LigRNA+ do not work in vitro and (3 out of 24) 12.5% of LigRNA- do not work in vitro. Are the authors able to glean any info on why so many LigRNA+ configurations fail? Could be useful for other aptamers/future designs.

8. There is a lot of text devoted to Figure S3 in the main text. I'm not sure that it's all necessary.

9. To test how decreased expression levels would affect the ligRNAs the authors use strong and weak promoters. However the expression of the gRNAs needs to be quantified. Also, are the results unexpected? Further, is this a generalizable phenomenon to all gRNAs or just the ligRNAs?

Reviewer #2:

Remarks to the Author:

Kundert et al. ("Controlling CRISPR-Cas9 with ligand-activated and 3 ligand-deactivated sgRNAs") describe results showing that RNA aptamers can be integrated with small gRNAs to generate ligand-responsive CRISPR-cas cleavage (in vitro) and CRISPRi transcriptional inhibition in *E. coli*. Although similar successes have been achieved in eukaryotic cells (Tang et al. 2017, Liu et al. 2016, cited here), to our knowledge, this work represents the first successful effort to directly control gRNA activity in *E. coli* in response to small-molecule inputs. The demonstration that the same general strategy can produce ligRNAs that either activate or repress gRNA activity is another noteworthy feature that, along with the apparent robustness to multiple spacer sequences, suggests that this strategy will be useful for others. Once the following concerns are addressed, it should be suitable for publication.

Major comments:

1. Overall, the manuscript would be strengthened by making stronger connections between the results of the rational design/in vitro assay and the in vivo FACS-seq based selection. As written, there are some hints that lessons learned from the design experiments in part 1 informed the library generation and selection in part 2. But, it is difficult to assess these claims. Supplementary

plots showing how the measured functions are related to the rational design variations should be presented alongside the current set of supplementary tables to make such comparisons possible.

2. It is unclear to this reviewer how the designs for the 3 different linking strategies were performed, and therefore whether or not the strategies were adequately explored. For example, very few induced dimerization constructs were built. As such, it is confusing that the different strategies are illustrated in figure 1 when they have so little impact on the rest of the paper.

3. Although figure 3 demonstrates the exciting possibility of multiplexed ligRNAs, the structural similarity of the Theo and 3MX aptamers diminishes enthusiasm that this is a general solution to the problem of designing ligand-controlled gRNAs. Providing another demonstration (even if only tested in vitro) that a more structurally-distinct aptamer can be assembled into ligRNAs would increase confidence in the generalizability of the approach.

4. It is interesting that the high-performing, rationally-designed devices show essentially no function in *E. coli*. Why? Co-transcriptional misfolding (which would be surprising given the demonstrated robustness to multiple spacer sequences)? Molecular crowding? Unexpected interactions with cellular proteins? Relatedly, can the authors at least speculate about why the ligRNAs are non-functional in *S. cerevisiae*?

Minor comments:

In figure 2g, please re-label which plot belongs to ligRNA +/-.

Point-by-point response and revisions # NCOMMS-18-14889-T

"Controlling CRISPR-Cas9 with ligand-activated and ligand-deactivated sgRNAs"

Reviewer #1 (Remarks to the Author):

Kundert et al. have developed a novel method to control S.pyogenes Cas9 activity via the use of small-molecule-responsive RNA aptamers. The authors use this platform to control Cas9 nuclease activity in vitro and to control Cas9-mediated repression at synthetic loci (GFP/RFP) in E.coli. The authors also show that this platform enables multiplexed control over repression of synthetic loci in E.coli. Overall, the technological platform is innovative and further it could have broad impact across multiple scientific disciplines. The work is also scientifically sound throughout. However, there are several opportunities to increase the impact of this work which would make the manuscript suitable for publication in Nature Communications. There are three general issues i) How generalizable is the technology to other spacers/species of Cas9 ii) Does the technology work at native loci in E.coli, and in other bacteria, iii) Why does the system fail in eukaryotic/mammalian cells. I have provided specific comments in connection to these issues below.

1. There appear to be relatively few spacer sequences tested in this manuscript, making it difficult to assess how this system would function across different targets. Unless I am mistaken, the total number of spacers tested in the main text is 28, broken down as follows:
Figure 1: 1 spacer in vitro (cutting) for aptamer designs.

Figure 2: 3 spacers in Ecoli (dCas9 repression) for design of Lig+ and Lig- designs

Figure 2: 24 spacers in vitro (cutting) for Lig+ and Lig- designs

Figure 3: 2 but Same spacers for (dCas9) repression In E.coli.

This seems like too few to generalize the programmability of the system. Targeting 1 or 2 endogenous loci in E.coli would not only help to address this, but would also increase impact via the nuclease-targeting of native, endogenous loci.

We have added new data targeting 4 endogenous loci in *E. coli* (new **Figures 3a,b** and **S9**). Text on page 10: "We next tested the ability of the ligRNAs to regulate endogenous genes (rather than the chromosomally integrated fluorescent proteins used in our screen) in two different bacterial species. We first tested 4 different spacers targeting the endogenous *lac* operon (*lacZ*, *lacI*, A-site and P-site) using a β -galactosidase assay (**Figure 3a**, **Figure S9**). ligRNA⁺ was functional at all loci, and ligRNA⁻ successfully targeted the two sites in *lacZ* and *lacI* (**Figure 3b**)."

2. Related to point 1, the following issues may decrease the applicability of this platform. Can the authors please comment on each to clarify.

A. For in vitro cleavage assays, all gRNA spacers begin with 3 Gs for "facile transcription by T7 polymerase".

This seems to bottleneck the gRNA search space from the outset of the study?

A: To clarify that this issue is not a significant bottleneck, we made the following changes:

Methods, in vitro DNA cleavage assay: "...and a spacer that began with 3 5' Gs (**Table S5**) for efficient transcription by T7 polymerase."

Caption to **Table S5**: "Note that for *in vitro* cleavage assays, sgRNAs were designed with 3 5' Gs for facile transcription by T7 polymerase. This would not represent a significant bottleneck for the sgRNA space in general as sgRNAs could also be designed with fewer 5' Gs, or

synthesized chemically without requirement for 5' Gs. Moreover, the positions at the 5' end of the spacer are the most tolerant to mismatches⁵."

B. The author's state that "for ligRNA⁻ constructs increased predicted affinity of the spacer for the aptamer sequence correlated with a smaller change in Cas9-mediated DNA cleavage in response to theophylline...This analysis suggested that spacers with predicted affinity for the aptamer could interfere with switching of the ligRNA⁻ function, providing a useful design criterion for functional spacers."

Doesn't this phenomenon also limit the design space for the protospacers that would be compatible with this system?

B: We agree with the reviewer that potential interactions between spacers and the theophylline aptamer could limit the space of sequences that could be targeted with our system. We have added on page 9:

"This analysis suggested that spacers with predicted affinity for the aptamer could interfere with switching of the ligRNA⁻ function, potentially limiting the space of sequences that could be targeted. Similar considerations also apply to the standard sgRNA scaffold, where internal pairing within the sgRNA sequence has been shown to affect CRISPR efficacy⁶. These considerations nonetheless provide useful design criteria for functional spacers."

C. The authors also comment that "spacers for this assay were chosen by a script that generated uniformly random sequences". Is this for all spacers used in this manuscript or only a certain subset?

Only spacers used in the *in vitro* cleavage assay in Figure 2h (supported by **Table S5**, **Figure S7** and **Figure S8**) were generated randomly. All other spacers were designed to target GFP and RFP or endogenous loci. We added the following clarification in the Methods section:

"Note that only spacers used in the *in vitro* cleavage assay were generated randomly. All other spacers were designed to target the respective genes (GFP, RFP, *folA*, *lacI*, *lacZ*) or genomic target sites as indicated."

3. Showing that the platform is extensible the TRACR portions of qRNAs from other bacterial species would substantially increase the impact of this manuscript and utility of this platform.
4. Especially given the importance that the authors place on being able to use this system in different bacteria. See "many species of bacteria do not have facile genetic controls available, and ligRNAs provide such controls with a minimal footprint. Moreover, temporally controlled gene expression programs are thought to be important for key biological processes in bacteria and ligRNAs provide a way to conduct large-scale screens..."

We added a new section and Figure panels (now **Figure 3c,d**) to the manuscript describing the use of the system in a different bacterial species:

"To determine whether ligRNAs also function in species other than *E. coli*, we created strains of *Pseudomonas aeruginosa* UCBPP-PA14 expressing both dCas9 and ligRNA⁺ targeting dihydrofolate reductase (DHFR). Repression of DHFR via CRISPRi lowers the minimal inhibitory concentration (MIC) of the antibiotic trimethoprim, which targets DHFR³⁹ (**Figure 3c**). As expected, the ligRNA⁺ strain consistently exhibited a lower MIC in the presence of theophylline (**Figure 3d**)."

We did not test other gRNA architectures using our ligRNAs as the mechanisms of ligRNA switching will be dependent on the specific sgRNA structure and its interaction with Cas9. We think our general approach is likely extensible but transfer to other sgRNA architectures would require repeating the screening process.

5. The platform does not work in eukaryotes/mammalian cells. This is a major limitation and is counter to many of the authors statements regarding the potential for broad applicability of this technology. For example “We envision that this method will be broadly useful for regulating essentially all applications of CRISPR-Cas9-mediated biological engineering”. I think that tempering some of that language is appropriate.

We agree and have made the following changes:

Introduction: “We envision that this method will be broadly useful for many applications of CRISPR-Cas9-mediated biological engineering in bacterial systems.”

We also focus the concluding paragraphs of the Results and Discussion on bacterial systems: “Nevertheless, there are already many useful applications for ligRNAs in bacteria. For example, many species of bacteria do not have facile genetic controls available, and ligRNAs provide such controls with a minimal footprint. Moreover, temporally controlled gene expression programs are thought to be important for key biological processes in bacteria⁸, and ligRNAs provide a way to conduct large-scale screens to probe these programs and their role in the interactions between bacteria and their environments⁹.”

6. Related to point #5 – two items in particular would be of tremendous value to the field and would strengthen this manuscript substantially. In order of importance:

A. Getting the system to work in mammalian cells (or even yeast) is critical.

We agree with the reviewer that function in eukaryotic cells would be a tremendous next development, but, as discussed with the editor, we would expect it to be a study of about equal magnitude as the current work.

B. Understanding why the system does not work in eukaryotes/mammalian cells would also be very useful to inform future studies.

The authors suggest that “While ligRNAs function robustly in bacteria, transferring them to eukaryotic systems will require further optimization which could be achieved using methods similar to our optimization (Figure 2) of the initial rational designs (Figure 1).4

I think that this would be a very important way to augment this work and potentially clarify the next steps in transferring this technology to eukaryotes. Especially since the authors are already well-versed in the assays/techniques in-house.

Some relevant and important items:

Are all the necessary components for this system appropriately expressed in eukaryotes?

Does chromatin affect activity?

Does the in vitro assay work with mammalian targets in mammalian cell lysates (i.e. is something inhibitory to efficacy in mammalian cells)?

We also agree that understanding the reasons for the lack of activity in eukaryotic systems is important, and we have added an expanded discussion. We reason that failure to express

Cas9, chromatin (while known to be an important factor for CRISPR efficacy) or inhibitory factors in eukaryotes are unlikely to be the main cause of the lack of transferability, because in each system (yeast and mammalian cells) we show that a control “wild-type” sgRNA produces efficient gene editing (mammalian cells) or moderate gene repression (*S. cerevisiae* experiments).

Instead, a key requirement for CRISPR/Cas9 to be controllable using ligand-responsive sgRNAs is that the system has to be limited by the concentration of active sgRNAs. We therefore believe a likely explanation for the failure of the system to function in eukaryotic cells is that active ligRNAs are not sufficiently close to the concentration required for activity (under the conditions tested). This hypothesis is supported by the observation that the ligRNA/Cas9 systems are inactive in eukaryotic cells in both the presence and the absence of theophylline. Under the same conditions, the wild-type sgRNA control system is active. It is likely that our ligRNA designs have decreased affinity for Cas9 as both ligRNA+ and ligRNA- constructs contain modifications in regions that are important for Cas9 binding, in particular the nexus. Therefore, we may not be reaching the required levels of active ligRNAs under our conditions.

Further, it is possible that the levels of active ligRNAs that we reach in *E. coli* under our conditions are higher, or, alternatively, that the levels required for function in *E. coli* are lower. (in either scenario, we selected controllable ligRNAs that function in the correct regime in *E. coli*).

[Redacted]

Taken together, we believe the best course of action is to adapt our *in vivo* selection platform to eukaryotic cells, which as mentioned above would essentially repeat the scope of the work.

To address the reviewer’s comment (as well as the questions of reviewer #2 below), we have changed the text as follows (page 11):

“While ligRNAs function robustly in bacteria, transferring them to eukaryotic systems will require further optimization. There are two key requirements for CRISPR/Cas9 to be controllable using ligand-responsive sgRNAs: First, the system has to be limited by the concentration of active sgRNAs, and second, this concentration has to change in response to ligand to reach sufficient levels for activity only in the “on” state. In contrast, we find that in eukaryotic cells the ligRNAs are inactive whether or not theophylline is present (**Figure S12, Figure S13**). This observation is unlikely due to the inability of theophylline to cross cell membranes, or to other general factors that may interfere with aptamer function, as the theophylline aptamer has been used to regulate gene expression in mammalian cells in multiple different contexts¹⁰. Moreover, it is also unlikely that normal CRISPR-Cas9 activity is inhibited in eukaryotes under our conditions, as control sgRNAs targeting the same sites are functional in our mammalian cell and *S. cerevisiae* experiments (**Figure S12, Figure S13**). One possible explanation for the lack of transferability to eukaryotic systems is that the ligRNAs have decreased affinity for Cas9 (as both ligRNAs contain mutations in regions that are important for Cas9 binding, the nexus in particular) and therefore do not reach the required active state concentration in the “on” state in eukaryotes. Future work might use screens similar to those described in **Figure 2** to develop suitable ligRNAs for eukaryotic systems.”

7. (9 out of 24) 37.5% of LigRNA+ do not work in vitro and (3 out of 24) 12.5% of LigRNA- do not work in vitro. Are the authors able to glean any info on why so many LigRNA+ configurations fail? Could be useful for other aptamers/future designs.

This is a valid point and we have added the following note to **Figure S8** showing the *in vitro* cleavage data:

“We note that a significant fraction of ligRNA+ constructs show low cleavage activities. Secondary structure prediction for our ligRNA+ constructs (**Figure S3**) suggest that they function by sequestering the critical nexus region through base-pairing with the aptamer sequence in the ligand-free inactive state. Ligand binding to the aptamer is then predicted to relieve this sequestration and stabilize a functional sgRNA conformation. To allow for complementary base-pairing with the aptamer in the inactive state, the selection experiment yielded slightly altered nexus sequences. As a consequence, the ligRNA+ constructs might have overall intrinsically lower affinity for Cas9, because the nexus makes critical interactions with the Cas9 protein. Future design efforts could add constraints to keep the nexus sequence as native-like as possible.”

8. There is a lot of text devoted to Figure S3 in the main text. I'm not sure that it's all necessary.

(Note: the original **Figure S3** is now **Figure S4**). We feel that this text adds to the manuscript by discussing and testing a potential mechanism for the initially unexpected result that ligRNA-inhibits (not activates) sgRNA function upon addition of theophylline. We are also substantially below the word limit for Nature Communications (5000 words for main text), so that inclusion of this section does not prevent us from including other information. We would thus prefer to keep the text in its current form (it is only 8 lines of extra text) for clarity.

9. To test how decreased expression levels would affect the ligRNAs the authors use strong and weak promoters. However the expression of the gRNAs needs to be quantified. Also, are the results unexpected? Further, is this a generalizable phenomenon to all gRNAs or just the ligRNAs?

The result that decreased ligRNA expression levels leads to decreased repression (as we observe in **Figure S5**) is expected, given that sgRNA concentrations are likely to be limiting under our conditions. In fact, it has been demonstrated recently by Carothers and coworkers [Fontana et al, ref 32 in the main text] for standard sgRNAs that the efficiency of CRISPRi gene repression decreases with decreasing sgRNA promoter strengths (under conditions of high Cas9 expression), therefore suggesting that the dependency of CRISPRi repression on sgRNA levels is general to both sgRNAs and our ligRNAs. We now cite the Carothers study (which used different strengths promoters but did not directly quantify sgRNA levels) and also directly quantify sgRNA levels under our conditions (new panel to **Figure S5**). Text changes are as follows (page 8):

“A recent study reported regulation of CRISPRi activity by modulating sgRNA expression levels in *E. coli*¹¹. To test how decreased expression levels would affect the ligRNAs, we replaced the strong constitutive promoter driving sgRNA expression (J23119) with a weak

constitutive promoter (J23150), confirmed decreased sgRNA expression by quantitative real-time polymerase chain reaction (qPCR) (**Figure S5a**), and repeated the CRISPRi assay.”

Reviewer #2 (Remarks to the Author):

Kundert et al. (“Controlling CRISPR-Cas9 with ligand-activated and 3 ligand-deactivated sgRNAs”) describe results showing that RNA aptamers can be integrated with small gRNAs to generate ligand-responsive CRISPR-cas cleavage (in vitro) and CRISPRi transcriptional inhibition in E. coli. Although similar successes have been achieved in eukaryotic cells (Tang et al. 2017, Liu et al. 2016, cited here), to our knowledge, this work represents the first successful effort to directly control gRNA activity in E. coli in response to small-molecule inputs. The demonstration that the same general strategy can produce ligRNAs that either activate or repress gRNA activity is another noteworthy feature that, along with the apparent robustness to multiple spacer sequences, suggests that this strategy will be useful for others. Once the following concerns are addressed, it should be suitable for publication.

Major comments:

1. Overall, the manuscript would be strengthened by making stronger connections between the results of the rational design/in vitro assay and the in vivo FACS-seq based selection. As written, there are some hints that lessons learned from the design experiments in part 1 informed the library generation and selection in part 2. But, it is difficult to assess these claims. Supplementary plots showing how the measured functions are related to the rational design variations should be presented alongside the current set of supplementary tables to make such comparisons possible.

We agree and have included a **Supplement Text** section and a new supplementary plot (**Figure S2**) to clarify and illustrate how the results from the rational designs informed the design of the libraries for FACS selection.

Supplemental Text:

“Influence of the rational designs on the FACS libraries

Although the rational designs only had weak ligand-sensitivity in a CRISPRi-based assay in *E. coli* (**Figure S1**), they informed the design of the libraries used in the FACS screens in four key aspects:

(i) Insertion sites: We used the same aptamer insertion sites as tested in the rational designs for the library designs, as we were able to find rational designs that were functional *in vitro* for all three sites (**Figure 1, Table S1**). **Figure S2** shows a comparison between the rational and library designs for each insertion site.

(ii) Strand displacement mechanism: Since the strand displacement was the most effective rational design strategy (**Table S1**), we built libraries consistent with sequestering the same motifs. In particular, the upper stem libraries (**Table S2: #1–#6**) included the bulge, and the second batch of hairpin libraries (**Table S2: #29, #30**) included part of the nexus. Note that secondary structure predictions of ligRNA+, which was isolated from library #29, are consistent with a strand displacement mechanism involving base-pairing between the aptamer (**Figure S3**).

(iii) Nexus libraries: The nexus stem is short and sensitive to mutation, both properties that make the strand displacement strategy difficult to implement. Instead, the rational designs with

the aptamer inserted into the nexus were designed using the stem replacement strategy. The designs included 0–5 nt linkers on either side of the aptamer, but only those with 2–5 nt linkers exhibited any *in vitro* cleavage activity (**Table S1**). The nexus libraries (**Table S2**, #7–#22) expanded on this approach, with randomized 2–5 nt linkers on either side of the aptamer. In addition, since inserting the aptamer into the nexus consistently produced sgRNAs that were deactivated (instead of activated) by theophylline, we screened all of the libraries for both ligand-activated and -deactivated ligRNAs.

(iv) Induced dimerization: Considering that the induced dimerization strategy was not successful *in vitro*, we did not pursue it in *E. coli*.”

New Figure S2: “Figure S2: Similarities between the rational designs and the FACS libraries. For each aptamer insertion site, the rational designs from **Figure 1d** (see also **Table S1**) are compared to the libraries listed in **Table S2**. The linker is defined as the sequence between the aptamer and the nearest region of the sgRNA scaffold shared between the designs and the libraries. Asterisks (*) indicate linkers in which not all nucleotides were mutated relative to the sgRNA scaffold.”

2. It is unclear to this reviewer how the designs for the 3 different linking strategies were performed, and therefore whether or not the strategies were adequately explored. For example, very few induced dimerization constructs were built. As such, it is confusing that the different strategies are illustrated in figure 1 when they have so little impact on the rest of the paper.

We apologize for not describing the linking strategies in more detail, and have now added an expanded description as Supplemental Text. We hope that the new sections (plus the section above on the relation to the FACS libraries) clarify the overall design strategy.

Supplemental Text:

“Additional details on the rational design strategies

Stem replacement (i): Relatively few of the interactions between Cas9 and its sgRNA are specific to the sequence of the sgRNA¹². Most of the interactions instead involve the RNA sugar-phosphate backbone, suggesting that in these regions the 3D shape of the sgRNA is the primary determinant of Cas9 binding. This hypothesis is supported by mutagenesis data¹³. As the theophylline aptamer adopts a conformation resembling duplex RNA only in the presence of ligand¹⁴, we reasoned that by replacing different portions of different stems with the aptamer, we might create ligand-activated sgRNAs. In some designs, we included sequences on either side of the aptamer following the pattern UUUCCC..., with the intention of discouraging the duplex state.

Induced dimerization (ii): Several aptamers, including the theophylline aptamer¹⁵, can be split and used to dimerize two strands of RNA in the presence of ligand¹⁶. Given that gRNAs in natural CRISPR systems comprise two strands of RNA (crRNA and tracrRNA) dimerized after a series of maturation steps, we reasoned that it might be possible to create ligand-activated gRNAs by splitting the gRNA into two non-functional halves and using a split-aptamer to artificially control their dimerization into a functional whole.

We applied this strategy to the upper stem, where the natural break between the crRNA and tracrRNA is located. We tested 5 different truncations of the upper stem, with the intent of eliminating any ligand-independent annealing while leaving enough of the upper stem for a functional Cas9/gRNA complex to form. None of the tested designs exhibited any cleavage with or without ligand (**Table S1**). Given the lack of detectable activity in *in vitro* conditions that

were relatively favorable (controllable concentrations of RNA, ligand, and Cas9; no competing interactions that could be present in cells), we did not pursue this strategy further.

Strand displacement (iii): Riboswitches commonly function by using an aptamer to switch between two conformational states with different base-pairing patterns. In one state, a functional motif is sequestered in an RNA duplex by a complementary strand. In the other state, the functional motif is revealed because the complementary strand base-pairs instead with another strand. This concept has been applied repeatedly to create artificial riboswitches¹⁷⁻¹⁹. We sought to apply the same concept to create ligand-activated sgRNAs. Specifically, we sought to conditionally sequester the regions of the sgRNA scaffold that are most sensitive to mutation¹³, reasoning that these regions would provide the most control over sgRNA function.

The regions we attempted to sequester were the bulge, the nexus, and the “ruler” (our name for the region between the nexus and the hairpin). The bulge forms several sequence-specific interactions with Cas9¹² and also introduces a necessary kink into the upper stem¹³. The nexus forms one sequence-specific interaction with Cas9¹² and a stem that is sensitive to changes in length¹³. The ruler is single-stranded, but sensitive to insertions or deletions, suggesting that the spacing between the nexus and the hairpin is important¹³.

We applied two topologies in our strand displacement designs (X is the sequence being sequestered, X' is a sequence complementary to X, X'' is a sequence complementary to X'):

X-tetraloop-X'-aptamer-X''

X-X''-aptamer-X'

In either case, our intention was that the active state would involve base-pairing between X' and X'', the inactive state would involve base-pairing between X' and X, and the aptamer would drive the transition from one state to the other. The aptamer was always inserted into one of the solvent-exposed stems: the upper stem, the nexus, or the hairpin. The particular sequences of X' and X'' were chosen to maintain all important features of the sgRNA as characterized by Briner et al.¹³, and depended on where the aptamer was inserted relative to the target sequence (**Table S1**).

Some of the above designs were either constitutively (i.e., both in the presence and in the absence of theophylline) active or inactive (**Table S1**). For the constitutively active designs, we attempted to weaken the active state by making complementary mutations in X' and X'' that would introduce wobble base-pairs or mismatches between X' and X. For the constitutively inactive designs, we attempted to weaken the inactive state by making mutations in X'' that would introduce wobble base-pairs or mismatches between X' and X''.

3. Although figure 3 demonstrates the exciting possibility of multiplexed ligRNAs, the structural similarity of the Theo and 3MX aptamers diminishes enthusiasm that this is a general solution to the problem of designing ligand-controlled gRNAs. Providing another demonstration (even if only tested in vitro) that a more structurally-distinct aptamer can be assembled into ligRNAs would increase confidence in the generalizability of the approach.

We have added additional data showing that ligRNAs can also be constructed using a thiamine pyrophosphate (TPP) aptamer. A new **Figure S11** shows that these ligRNAs function in *E. coli*. We have added a section describing these results on pages 10-11:

“Going further, we also created thiamine-responsive ligRNAs by repeating our FACS screens (**Figure 2**) with libraries containing the thiamine pyrophosphate aptamer (**Figure S11, Table S2, Table S3**). Although these ligRNAs have a narrower dynamic range (6-fold) than the

theophylline or 3-methylxanthine ligRNAs, they demonstrate that our overall strategy for creating ligRNAs is applicable to other ligands.”

4. It is interesting that the high-performing, rationally-designed devices show essentially no function in *E. coli*. Why? Co-transcriptional misfolding (which would be surprising given the demonstrated robustness to multiple spacer sequences)? Molecular crowding? Unexpected interactions with cellular proteins? Relatedly, can the authors at least speculate about why the ligRNAs are non-functional in *S. cerevisiae*?

To address the first set of questions (performance in *E. coli*), we have added an additional discussion to the caption of **Figure S1** that shows data testing the high-performing rational designs in *E. coli*:

“Differences between *in vitro* and *in vivo* sgRNA activities have been reported in other systems²⁰. One possible explanation is misfolding *in vivo*, which is alleviated *in vitro* because of an extra refolding step. This scenario could explain the case where constructs that were active *in vitro* were mostly inactive in *E. coli* irrespective of the presence of the ligand (#24, #25, #61). We also observe the case where a construct that is active and switchable *in vitro* is also active (represses) in *E. coli*, but unresponsive to ligand (#3, #80, #85). In this case, it is likely that the concentration of active ligRNA is above the sensitive range in both presence and absence of the ligand.”

Relatedly, and as outlined above in response to comments from reviewer #1, we speculate that the concentration of the active ligRNA species, relative to their affinities for Cas9, are likely outside the sensitive range in *S. cerevisiae*. Please see our response to point 6 raised by reviewer #1.

Minor comments:

In figure 2g, please re-label which plot belongs to ligRNA+/-.

We have made this change in the Figure (it is now Figure 2 panel h).

We would like to thank the reviewers again for their comments that improved our manuscript, and hope it is now suitable for publication in Nature Communications.

References cited the response letter (note numbering is different in Manuscript/Supplement):

1. Bernstein, J.A., Khodursky, A.B., Lin, P.H., Lin-Chao, S. & Cohen, S.N. Global analysis of mRNA decay and abundance in Escherichia coli at single-gene resolution using two-color fluorescent DNA microarrays. *Proc Natl Acad Sci U S A* **99**, 9697-702 (2002).
2. Singh, D., Sternberg, S.H., Fei, J., Doudna, J.A. & Ha, T. Real-time observation of DNA recognition and rejection by the RNA-guided endonuclease Cas9. *Nat Commun* **7**, 12778 (2016).
3. Knight, S.C., Xie, L., Deng, W., Guglielmi, B., Witkowsky, L.B., Bosanac, L., Zhang, E.T., El Beheiry, M., Masson, J.B., Dahan, M., Liu, Z., Doudna, J.A. & Tjian, R. Dynamics of CRISPR-Cas9 genome interrogation in living cells. *Science* **350**, 823-6 (2015).
4. Sternberg, S.H., Redding, S., Jinek, M., Greene, E.C. & Doudna, J.A. DNA interrogation by the CRISPR RNA-guided endonuclease Cas9. *Nature* **507**, 62-7 (2014).
5. Fu, Y., Sander, J.D., Reyon, D., Cascio, V.M. & Joung, J.K. Improving CRISPR-Cas nuclease specificity using truncated guide RNAs. *Nat Biotechnol* **32**, 279-284 (2014).
6. Horlbeck, M.A., Gilbert, L.A., Villalta, J.E., Adamson, B., Pak, R.A., Chen, Y., Fields, A.P., Park, C.Y., Corn, J.E., Kampmann, M. & Weissman, J.S. Compact and highly active next-generation libraries for CRISPR-mediated gene repression and activation. *Elife* **5**(2016).
7. Peters, J.M., Koo, B.-M., Patino, R., Heussler, G.E., Hearne, C.C., Inclan, Y., Hawkins, J.S., Lu, C.H.S., Harden, M.M., Osadnik, H., Peters, J.E., Engel, J.N., Dutton, R.J., Grossman, A.D., Gross, C.A. & Rosenberg, O.S. Mobile-CRISPRi: Enabling Genetic Analysis of Diverse Bacteria. *bioRxiv* (2018).
8. Laub, M.T., McAdams, H.H., Feldblyum, T., Fraser, C.M. & Shapiro, L. Global analysis of the genetic network controlling a bacterial cell cycle. *Science* **290**, 2144-8 (2000).
9. Peters, J.M., Colavin, A., Shi, H., Czarny, T.L., Larson, M.H., Wong, S., Hawkins, J.S., Lu, C.H.S., Koo, B.M., Marta, E., Shiver, A.L., Whitehead, E.H., Weissman, J.S., Brown, E.D., Qi, L.S., Huang, K.C. & Gross, C.A. A Comprehensive, CRISPR-based Functional Analysis of Essential Genes in Bacteria. *Cell* **165**, 1493-1506 (2016).
10. Berens, C., Groher, F. & Suess, B. RNA aptamers as genetic control devices: The potential of riboswitches as synthetic elements for regulating gene expression. *Biotechnology Journal* **10**, 246-257 (2015).
11. Fontana, J., Dong, C., Ham, J.Y., Zalatan, J.G. & Carothers, J.M. Regulated Expression of sgRNAs Tunes CRISPRi in E. coli. *Biotechnol J* **13**, e1800069 (2018).
12. Nishimasu, H., Ran, F.A., Hsu, P.D., Konermann, S., Shehata, S.I., Dohmae, N., Ishitani, R., Zhang, F. & Nureki, O. Crystal structure of Cas9 in complex with guide RNA and target DNA. *Cell* **156**, 935-49 (2014).
13. Briner, A.E., Donohoue, P.D., Gooma, A.A., Selle, K., Slorach, E.M., Nye, C.H., Haurwitz, R.E., Beisel, C.L., May, A.P. & Barrangou, R. Guide RNA functional modules direct Cas9 activity and orthogonality. *Mol Cell* **56**, 333-9 (2014).
14. Zimmermann, G.R., Jenison, R.D., Wick, C.L., Simorre, J.P. & Pardi, A. Interlocking structural motifs mediate molecular discrimination by a theophylline-binding RNA. *Nat Struct Biol* **4**, 644-9 (1997).
15. Jiang, H., Ling, K., Tao, X. & Zhang, Q. Theophylline detection in serum using a self-assembling RNA aptamer-based gold nanoparticle sensor. *Biosensors and Bioelectronics* **70**, 299-303 (2015).
16. Chen, A., Yan, M. & Yang, S. Split aptamers and their applications in sandwich aptasensors. *TrAC Trends in Analytical Chemistry* **80**, 581 - 593 (2016).

17. Lynch, S.A. & Gallivan, J.P. A flow cytometry-based screen for synthetic riboswitches. *Nucleic Acids Res* **37**, 184-92 (2009).
18. Muranaka, N., Abe, K. & Yokobayashi, Y. Mechanism-guided library design and dual genetic selection of synthetic OFF riboswitches. *Chembiochem* **10**, 2375-81 (2009).
19. Werstuck, G. & Green, M.R. Controlling gene expression in living cells through small molecule-RNA interactions. *Science* **282**, 296-8 (1998).
20. Thyme, S.B., Akhmetova, L., Montague, T.G., Valen, E. & Schier, A.F. Internal guide RNA interactions interfere with Cas9-mediated cleavage. *Nat Commun* **7**, 11750 (2016).

Reviewers' Comments:

Reviewer #1:

Remarks to the Author:

I am impressed by the authors' new data, explanations, and agreeable responses to both reviewers' input. I believe that the authors' revisions have strengthened this manuscript, and further that their work herein will produce broad impacts and widespread value to the scientific community. The manuscript is scientifically sound, important, and exciting. I anticipate that it will be of high interest and usefulness to all biomedical researchers and readers of Nature Communications. The manuscript is well-suited for publication in Nature Communications without further revision.

Reviewer #2:

Remarks to the Author:

I have reviewed the revision rebuttal and have no further concerns. It is a nice piece of work that will be welcomed by the community -- and is suitable for publication.

Response to reviewers' comments

The reviewers raised no further issues (their comments are copied below).

REVIEWERS' COMMENTS:

Reviewer #1 (Remarks to the Author):

I am impressed by the authors' new data, explanations, and agreeable responses to both reviewers' input. I believe that the authors' revisions have strengthened this manuscript, and further that their work herein will produce broad impacts and widespread value to the scientific community. The manuscript is scientifically sound, important, and exciting. I anticipate that it will be of high interest and usefulness to all biomedical researchers and readers of Nature Communications. The manuscript is well-suited for publication in Nature Communications without further revision.

Reviewer #2 (Remarks to the Author):

I have reviewed the revision rebuttal and have no further concerns. It is a nice piece of work that will be welcomed by the community -- and is suitable for publication.